# A Diffusion Model for Regular Time Series Generation from Irregular Data with Completion and Masking

**Gal Fadlon**[*]  **Idan Arbiv**[*]  **Nimrod Berman**  **Omri Azencot**
Department of Computer Science
Ben-Gurion University of The Negev
{galfad, arbivid, bermann}@post.bgu.ac.il
azencot@bgu.ac.il

## Abstract

Generating realistic time series data is critical for applications in healthcare, finance, and science. However, irregular sampling and missing values present significant challenges. While prior methods address these irregularities, they often yield sub-optimal results and incur high computational costs. Recent advances in regular time series generation, such as the diffusion-based ImagenTime model, demonstrate strong, fast, and scalable generative capabilities by transforming time series into image representations, making them a promising solution. However, extending ImagenTime to irregular sequences using simple masking introduces "unnatural" neighborhoods, where missing values replaced by zeros disrupt the learning process. To overcome this, we propose a novel two-step framework: first, a Time Series Transformer completes irregular sequences, creating natural neighborhoods; second, a vision-based diffusion model with masking minimizes dependence on the completed values. This approach leverages the strengths of both completion and masking, enabling robust and efficient generation of realistic time series. Our method achieves state-of-the-art performance, achieving a relative improvement in discriminative score by $70\%$ and in computational cost by $85\%$. Code is at https://github.com/azencot-group/ImagenI2R.

## 1 Introduction

Time series data is essential in fields such as healthcare, finance, and science, supporting critical tasks like forecasting trends, detecting anomalies, and analyzing patterns [18, 24, 34]. Beyond direct analysis, generating synthetic time series has become increasingly valuable for creating realistic proxies of private data, testing systems under new scenarios, exploring "what-if" questions, and balancing datasets for training machine learning models [6, 40]. The ability to generate realistic sequences enables deeper insights and robust applications across diverse domains.

In practice, however, time series is often *irregular*, with unevenly spaced measurements and missing values. These irregularities arise from limitations in data collection processes, such as sensor failures, inconsistent sampling, or interruptions in monitoring systems [28, 44]. This irregularity poses a unique challenge for generating regular time series—where intervals are consistent, and the data follows the same distribution as if it were regularly observed [26]. The main goal of this paper is to generate regular sequences by training on irregularly-sampled time series using deep neural networks.

The synthesis of regular time series from irregular ones is a fundamental challenge, yet existing approaches remain scarce, with notable examples being GT-GAN and KoVAE [26, 39]. Unfortunately, these methods suffer from several limitations. First, they rely on generative adversarial networks (GANs) and variational autoencoders (VAEs), which have recently been surpassed in performance

---

[*]Equal Contribution

39th Conference on Neural Information Processing Systems (NeurIPS 2025).

by diffusion-based tools [11, 59, 38]. Second, both GT-GAN and KoVAE utilize a computationally-demanding preprocessing step based on neural controlled differential equations (NCDEs) [28], rendering these methods impractical for long time series. For instance, KoVAE requires $\approx 6.5\times$ more training time in comparison to our approach (See Fig. 3). Third, these methods inherently assume that the data, completed by NCDE, accurately reflects the true underlying distribution, which can introduce catastrophical errors when this assumption fails. In particular, their performance lags far behind that of models trained on regular time series, with state-of-the-art results on irregular discriminative benchmarks being, on average, $540\%$ worse than those on regular benchmarks.

To address these shortcomings, we base our approach on a recent diffusion model for time series, *ImagenTime* [38]. This method maps time series data to images, enabling the use of powerful vision-based diffusion neural architectures. Leveraging a vision-based diffusion generator offers a significant advantage: regular time series can be generated from irregular ones using a straightforward masking mechanism. Specifically, missing values in the series are seamlessly ignored during the denoising process in training, akin to techniques used in image inpainting tasks [37, 12].

However, while this straightforward masking approach is simple and achieves strong results (see Tab. 7), we identify a significant limitation. Missing values in the time series are mapped to zeros in the image, resulting in "unnatural" neighborhoods that mix valid and invalid information. This can pose challenges for diffusion backbones, such as U-Nets with convolutional blocks, where the convolution kernels are not inherently masked and may inadvertently propagate errors from these artificial neighborhoods. To address this issue, we propose a two-step generation process. In the first step, we complete the irregular series using our adaptation of an efficient Time Series Transformer (TST) approach [61], significantly reducing computational overhead and enabling the generation of long time series. In the second step, we apply the straightforward masking approach described earlier. Crucially, this combination of completion and masking allows the model to learn from "natural" image neighborhoods while mitigating the reliance on fully accurate completed information through the use of masked loss minimization. Overall, our approach, built on a vision diffusion backbone, enables effective modeling of long time series while making minimal assumptions about pre-completed data, resulting in significantly efficient and improved generation performance.

We conduct a comprehensive evaluation of our approach on standard irregular time series benchmarks, benchmarking it against state-of-the-art methods. Our model consistently demonstrates superior generative performance, effectively bridging the gap between regular and irregular settings. Furthermore, we extend the evaluation to medium-, long- and ultra-long-length sequence generation, assessing performance across 12 datasets and 12 tasks. The results highlight the robustness and efficiency of our method, achieving consistent improvements over existing approaches. Our key contributions are summarized as follows:

1. We introduce a novel generative model for irregularly-sampled time series, leveraging vision-based diffusion approaches to efficiently and effectively handle sequences ranging from short to long lengths.

2. In contrast to existing methods that assume completed information is drawn from the data distribution, we treat it as a weak conditioning signal and directly optimize on the observed signal using a masking strategy.

3. Our approach achieves state-of-the-art performance across multiple generative tasks, delivering an average improvement of $70\%$ in discriminative benchmarks while reducing computational requirements by $85\%$ relative to competing methods.

## 2 Related Work

**Diffusion models** [49] have recently demonstrated groundbreaking generative capabilities, surpassing VAEs and GANs [29, 17] primarily on image generation [21, 14, 45]. The immense success of diffusion-based approaches has spurred a wave of recent advancements, encompassing both theoretical developments [50, 35] and practical applications [37, 22, 30, 4, 5, 3].

**Generative modeling of time series** is an emerging field, with pioneering approaches predominantly relying on GANs [58, 33, 31] and VAEs [13, 39]. Recently, inspired by the success of diffusion models, there has been a growing trend to adapt these techniques to various time series tasks [53, 43], including generative modeling [11, 59, 16]. Notably, the ImagenTime approach [38] has achieved state-of-the-art performance on regular generative tasks for sequences of varying lengths, from short

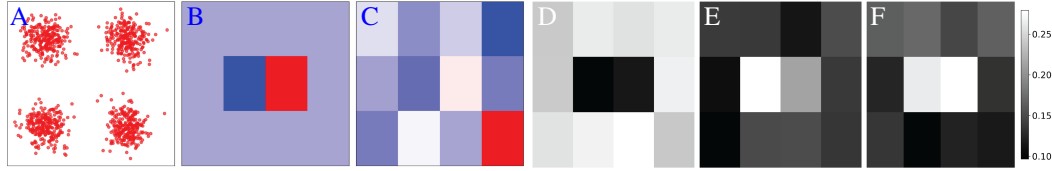

score = 0.71    score = 0.67    score = **0.32**

Figure 1: A data point (A) is mapped to an image with zeros and the coordinates in the center (B). Denoising the entire image yields inferior kernels (D) in comparison to masking (E). Constructing natural neighborhoods (C), yields consistent kernels and better scores (F).

to very long, by transforming time series into images and leveraging vision-based diffusion backbones. Our work builds on ImagenTime, extending it to address the challenging setting of generating regular time series information from irregularly-sampled data.

**Irregular time series** modeling has been a longstanding task. Modern machine learning methods have made significant strides by framing the problem through the lens of differential equations [46, 28]. Subsequent efforts have explored alternative architectures, including recurrent neural networks [47] and transformers [61, 9]. However, learning from irregular sequences has received comparatively less attention, with notable contributions such as GT-GAN and KoVAE [26, 39], both relying on NCDE [28]. Despite their promise, NCDE-based methods are costly during preprocessing and training, limiting the applicability of GT-GAN and KoVAE in handling long time series. Further, replacing NCDE by efficient components such as TST [61], yields suboptimal results (see Tab. 7).

## 3  Background

**Problem statement.**   We learn the underlying distribution of time series data from irregularly sampled observations and generating regular time series from it. Specifically, given a set of irregularly sampled sequences, our goal is to learn a model that approximates the true data distribution $p_{\text{data}}(x_{1:T})$ and enables sampling of complete time series $x_{1:T}$ from the learned distribution $p_\theta(x_{1:T})$. Formally, we consider a dataset of irregularly sampled time series, represented as $\{x^j_{t_1:t_n}\}_{j=1}^N$, where each sequence consists of observations at non-uniform time steps $t_1 : t_n = [t_1, t_2, \ldots, t_n]$ with $t_1 \geq 1$ and $t_n \leq T$. The challenge is to leverage these incomplete sequences to model the full distribution and generate realistic, regularly sampled time series that align with the true data distribution.

**ImagenTime** employs the delay embedding transformation to map time series data into images, enabling their processing with powerful vision-based diffusion models [38]. Given an input multivariate regular time series $x_{1:T} \in \mathbb{R}^{d \times T}$ with $d$ features and length $T$, the delay embedding constructs an image $x_{\text{img}} \in \mathbb{R}^{d \times w \times h}$ by placing $x$'s values over the columns of $x_{\text{img}}$ per channel, where $w, h$ are user-defined parameters. During training, noise is added to the image $x_{\text{img}}$ at different timesteps, forming $x_{\text{img}}(t)$. The diffusion model, parameterized by $s_\theta$, learns to denoise these images by approximating the score function $s_\theta(x_{\text{img}}, t)$. Inference begins with a noise sample $x_{\text{img}}(T) \sim \mathcal{N}(0, I)$. This sample is iteratively denoised using the learned score function to produce a cleaned image $x_{\text{img}}(0)$. The inverse delay embedding transform is then applied to $x_{\text{img}}(0)$, reconstructing the original time series $\tilde{x}_{1:T}$. Importantly, the inverse transformation of delay embedding is inherently non-unique, as the time series values are repeated within the image representation, suggesting various designs can be considered as we discuss in Sec. 4. Finally, a crucial advantage of ImagenTime is its effectivity in handling long series, e.g., a time series of length $65k$ is transformed to an image of size $256 \times 256$.

**TST** leverages the self-attention mechanism of Transformers to model temporal dependencies and long-range interactions in time-series data effectively. Unlike traditional sequence models such as RNNs or LSTMs, TST processes the entire sequence simultaneously, enabling parallelism and mitigating the vanishing gradient problem. The architecture includes input projection to a higher-dimensional feature space, positional encodings to capture temporal order, and a stack of Transformer encoder layers with flexible normalization and activation options. TST is particularly suitable for imputation tasks, as it can handle irregularly-sampled data and missing values through explicit masking and preprocessing. Additionally, its self-attention mechanism inherently supports long sequences, making it robust for capturing global context and dependencies in extended time-series data. By eliminating the need for computationally expensive preprocessing techniques, such as calculating coefficients for cubic splines or other interpolation methods, TST achieves significant speed advantages while providing a scalable, efficient, and accurate solution for tasks like forecasting, classification, anomaly detection, and imputation.

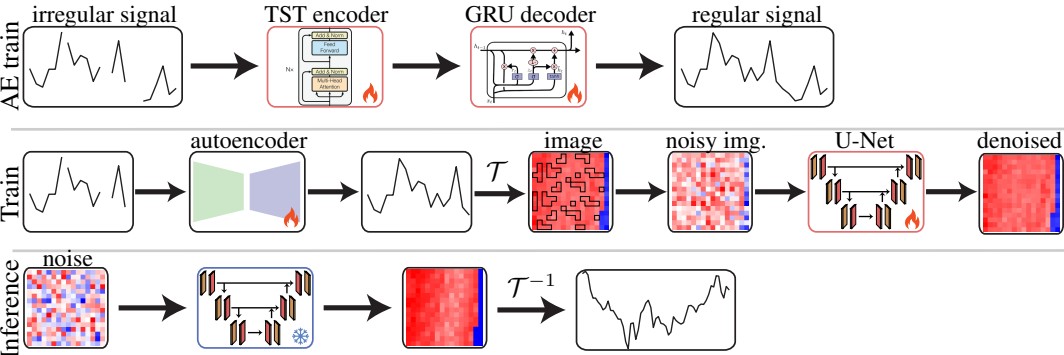

Figure 2: In the first step (top), we train a TST-based autoencoder, which we use during the second step (middle), where a vision diffusion model is trained with masking over non-active pixels. Inference (bottom) is done similarly to ImagenTime.

## 4 Method

While ImagenTime does not address the challenge of irregularly-sampled time series, a simple extension can enable it to generate regularly-sampled time series by training on irregular data. The key idea involves employing a *mask* during the loss computation. This mask ensures that only "active" pixels–those corresponding to observed time series values–are considered in the loss calculation, while "non-active" pixels, representing missing information, are effectively ignored. This approach enables effective learning from incomplete data while preserving the integrity of the observed information, offering two key advantages: (i) the mask is architecture-agnostic, making it compatible with any diffusion backbone, and (ii) the inference procedure of ImagenTime remains entirely unchanged.

### 4.1 Unnatural image neighborhoods

Unfortunately, the straightforward approach has a fundamental limitation: although non-active pixels are ignored during loss computation, they are still processed by the network. In practice, missing values are replaced with zeros, resulting in "unnatural" pixel neighborhoods. Specifically, while zeros may occasionally occur in non-zero segments of a time series, their repeated presence is highly unlikely, leading to inconsistencies. In other words, masking is not applied at the architecture level, potentially hindering the effective learning of neural components.

To demonstrate this phenomenon, we consider the following toy experiment. We generate $1000$ two-dimensional points, drawn from a multivariate Gaussian distribution with four centers (Fig. 1A). Given this distribution, we create an irregular dataset of $3 \times 4$ images, $\mathcal{S}_{\text{irregular}}$, by taking a data point, setting all pixels to zero except those at the center, corresponding to the $x$ and $y$ coordinates of the original point (Fig. 1B). Then, we train two diffusion models to: (i) predict the score across the entire image, and (ii) predict only the two central pixels via masking.

We evaluate the models by comparing their score estimation loss only on the two central pixels, regardless of the training strategy. Our results indicate that masking does *not* improve score estimation, yielding scores of $0.71$ vs. $0.67$ for Setups (i) and (ii), respectively. In addition, we also inspect the convolution kernels by averaging across channels the $L_1$ norm of each spatial position (Fig. 1D,E). As can be seen, the kernel in Setup (i) heavily attends to zero-valued pixels instead of focusing on the essential central pixels, suggesting that "unnatural" neighbors may be detrimental. In contrast, the masked kernel from Setup (ii) largely ignores non-relevant zeros and prioritizes the middle pixels.

### 4.2 Our approach

One possible solution to alleviate the phenomenon of unnatural neighborhoods is to implement masking at the kernel level, but this would require modifications tailored to each neural architecture, thereby restricting the approach's flexibility and its straightforward applicability across different models. For instance, while our work employs a convolutional U-Net, recent transformer-based architectures have emerged as highly effective diffusion backbones [41]. Accommodating such diversity in architectures would require a more generalized solution.

To construct more natural pixel neighborhoods while remaining architecture-agnostic, we take inspiration from the two-stage pipelines of GT-GAN and KoVAE [26, 39]. Our method likewise uses a two-step training scheme. First, we complete missing values in irregularly sampled time series using

TST [61] to obtain a regularly sampled sequence. Second, we transform the completed series into an image and apply denoising as in ImagenTime, with one key change: during the loss computation we *mask* the pixels that originated from completion (see App. B), following the straightforward masking strategy discussed earlier. In the toy experiment of Sec. 4.1, the completed neighborhood (Fig. 1C) enables learning of consistent kernels (Fig. 1F) and improves score estimation to $0.32$.

We also replicated the synthetic experiment on a real-world **Stocks** dataset, using a larger convolutional kernel and comparing the two ways to complete irregular neighborhoods: (i) zero filling and (ii) natural-neighbor filling. The Stocks results mirrored the synthetic study, reinforcing our hypothesis that "unnatural" neighbors (e.g., zeros) are detrimental. Evaluated by score-estimation loss on active pixels, simple masking delivered only a marginal gain ($0.81 \rightarrow 0.79$). Visual inspection likewise matched the synthetic patterns. In contrast, our proposed approach, which avoids zero padding and learning from semantically valid neighborhoods, significantly improved performance, reducing the loss to **0.29** and yielding more consistent kernel behavior. See App.E.1 for visuals.

This combination of *completion + masking* addresses the two primary challenges of irregular sequences. Completion creates natural neighborhoods so convolutional kernels learn from values closer to the true data distribution; masking prevents over-reliance on imputed values by excluding them from the loss, striking a balance between leveraging and mitigating incomplete information. Figure 2 illustrates our pipeline: autoencoder pretraining (top), main training (middle), and inference (bottom). $\mathcal{T}$ and $\mathcal{T}^{-1}$ denote the delay-embedding transform and its inverse; the fire and snowflake icons indicate trainable and frozen modules, respectively.

Importantly, we identify limitations with $\mathcal{T}^{-1}$ that was proposed in [38]. Specifically, in their approach, only the first pixel corresponding to each time series value is used for reconstruction. Specifically, if $x_i$ is mapped to multiple image indices, the original method selects the first corresponding pixel in the image for reconstruction. We modify this inverse transformation by aggregating information from all corresponding image indices and computing the average of the associated pixels for each $x_i$. For a given $x_{1:L} \in \mathbb{R}^L$, both methods ensure that $f^{-1}(f(x)) = x$. See Sec. 5.6 for an ablation study of these two methods.

# 5 Results

## 5.1 Quality vs. Complexity

We first compare the discriminative score and training time of our method against KoVAE across different sequence lengths $(24, 96, 768)$. Both models were evaluated under identical conditions, utilizing the same GPU and batch size to ensure a fair comparison. Training time was measured until each model converged to its best result. The discriminative score and training time were averaged over all missing rates and datasets for each sequence length and each method separately. As illustrated in Fig. 3 and Tab. 1, our approach achieves an average speedup of approximately **6.5**× and an average improvement of about **3.4**× in the discriminative score compared to KoVAE. The results demonstrate that our method not only trains significantly faster but also generates data that more closely resembles the real distribution compared to KoVAE. Full results can be seen in App. E.7

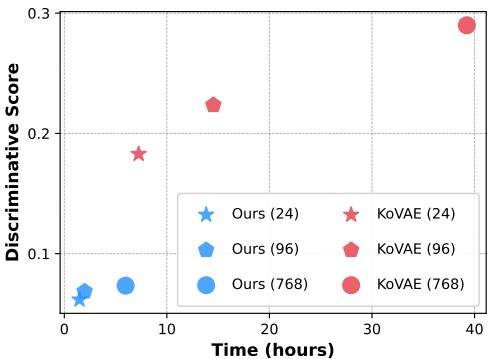

Figure 3: Discriminative score vs. training time for our approach and KoVAE across different lengths $(24, 96,$ and $768)$. Lower discriminative scores and shorter training times are better.

## 5.2 Quantitative Evaluation

Our quantitative evaluations assess missing rate setups of 30%, 50%, and 70%. For example, in the 30% missing rate case, we randomly omit 30% of the data in each training sample. Additionally, we extend the standard benchmark, which typically considers a sequence length of 24, to include longer sequences of 96, 768, and 10,920, providing a more comprehensive evaluation across varying temporal scales. We utilize a diverse set of datasets, extending beyond common benchmarks such as Sine, Stock, Energy, and MuJoCo to include additional real-world datasets: ETTh1, ETTh2, ETTm1, ETTm2, Weather, Electricity, KDD-Cup, and Traffic. We compare against the popular TimeGAN

approach [58], adapted to handle irregular data by incorporating the time difference between samples as input. In addition, we also consider the recent GT-GAN [26] and KoVAE [39].

We evaluate the performance of our model using the *discriminative* and *predictive* tasks suggested by [58]. In the discriminative task, we measure the similarity between real and synthetic samples by training a classifier to distinguish between the two, reporting $|0.5 - \text{acc}|$, where acc is the accuracy of the classifier. For the predictive task, we adopt the "train on synthetic, test on real" protocol, where a predictor is trained on synthetic data and tested on real data. The performance is evaluated using the Mean Absolute Error (MAE). We also consider irregular time series metrics: the *Context-FID score* [25], which quantifies the similarity in distribution between synthetic and real data, and the *correlation score* [33], which evaluates the feature-level relationship between the two datasets. The Context-FID score is computed by encoding both synthetic and real sequences using TS2Vec [60] and calculating the FID score on the representations. The correlation score measures the covariance of features between real and synthetic data, with a focus on assessing their alignment. Full details are provided in App. D.3. For all metrics, lower scores are better.

Tab. 2 details the benchmark results for a sequence length of 24. The values represent averages over the $30\%, 50\%$ and $70\%$ missing rates, where the full results are provided in Tab. 18. In general, our approach presents dramatic improvements across all metrics with respect to the second-best approach (typically KoVAE). Following [39], we define the relative improvement error as $e_{\text{rel}} = (e_2 - e_1)/e_2$, where $e_2$ is the second-best error and $e_1$ is ours. In this metric, averaged across all datasets, our method improves by **74.2%, 15.0%, 78.5%, 62.1%** in the discriminative, predictive, context-FID, and correlation scores, respectively. We also compared our approach to KoVAE in the medium (96) and long (768) lengths on all datasets excluding MuJoCo and electricity. The results appear in Tab. 3,

showing the superiority of our method and its advantages to longer horizons, where we achieved mean relative improvement of **72.6%, 29.7%, 92.1%, 73.25%** in the discriminative, predictive, context-FID, and correlation scores. See the full results in Tabs. 19, 20. Finally, for ultra-long sequences (10,920) on the KDD-Cup dataset, our model showed improvements of **36.6%** in the discriminative score (see Tab. 4), showing its ability to generate realistic synthetic data even at extreme sequence lengths.

Table 4: Discriminative results of ultra-long (10,920) sequences on KDD-Cup for different missing rates.

| Method | 30% | 50% | 70% |
|--------|-----|-----|-----|
| KoVAE | 0.375 | 0.410 | 0.499 |
| Ours | **0.155** | **0.288** | **0.392** |

### 5.3 Qualitative Evaluation

We evaluate the similarity between the generated sequences and the real data using qualitative metrics. Specifically, we employ two visualization techniques [58]: (i) projecting the real and synthetic data into a two-dimensional space using t-SNE [55], and (ii) performing kernel density estimation to visualize the corresponding probability density functions (PDFs). Fig. 4 illustrates these results for the 70% missing rate setting over various datasets and sequence lengths: Energy (24), Weather (96), and Stocks (768). The top row shows the two-dimensional point clouds of real (blue), KoVAE (green), and our data obtained via t-SNE, while the bottom row displays their respective PDFs. Overall, our approach demonstrates strong alignment in both visualizations. In the t-SNE plots (top row), a high degree of overlap between real and synthetic samples is observed. Similarly, in the PDF plots (bottom row), the trends and behaviors of the distributions are closely aligned. For additional results, including the regular and irregular 30% and 50% settings, please refer to App. E.6.

Table 1: Train time (hours) for lengths (24, 96, 768), averaged over missing rates (30%, 50%, 70%).

| | Model | ETTh1 | ETTh2 | ETTm1 | ETTm2 | Weather | Electricity | Energy | Sine | Stock | Mujoco |
|---|-------|-------|-------|-------|-------|---------|-------------|--------|------|-------|--------|
| **24** | GT-GAN | 7.44 | 3.68 | 5.14 | 3.60 | 5.04 | 6.45 | 5.25 | 4.09 | 3.15 | 2.17 |
| | KoVAE | 6.49 | 10.79 | 10.14 | 6.55 | 5.840 | 16.57 | 9.31 | 5.59 | 2.04 | 1.15 |
| | Ours | **1.28** | **2.76** | **0.96** | **1.43** | **3.66** | **2.44** | **1.00** | **0.48** | **0.21** | **0.60** |
| **96** | KoVAE | 19.70 | 10.82 | 14.61 | 26.51 | 22.06 | - | 13.68 | 10.76 | 6.45 | - |
| | Ours | **1.52** | **1.90** | **1.24** | **1.85** | **1.87** | - | **1.72** | **1.46** | **0.76** | - |
| **768** | KoVAE | 31.53 | 37.66 | 67.97 | 72.58 | 52.93 | - | 21.04 | 14.64 | 16.27 | - |
| | Ours | **5.38** | **2.61** | **12.20** | **7.49** | **9.47** | - | **4.96** | **8.24** | **2.74** | - |

Table 2: Averaged results over 30%, 50%, 70% missing rates for length 24. Lower values are better.

| | Model | ETTh1 | ETTh2 | ETTm1 | ETTm2 | Weather | Electricity | Energy | Sine | Stock |
|---|---|---|---|---|---|---|---|---|---|---|
| **Disc.** | TimeGAN-$\Delta t$ | 0.499 | 0.499 | 0.499 | 0.499 | 0.497 | 0.499 | 0.474 | 0.497 | 0.479 |
| | GT-GAN | 0.471 | 0.369 | 0.412 | 0.366 | 0.481 | 0.427 | 0.325 | 0.338 | 0.249 |
| | KoVAE | 0.197 | 0.081 | 0.050 | 0.067 | 0.332 | 0.498 | 0.323 | 0.043 | 0.118 |
| | Ours | **0.037** | **0.009** | **0.012** | **0.011** | **0.057** | **0.384** | **0.080** | **0.010** | **0.008** |
| **Pred.** | TimeGAN-$\Delta t$ | 0.267 | 0.336 | 0.235 | 0.314 | 0.394 | 0.262 | 0.457 | 0.334 | 0.072 |
| | GT-GAN | 0.186 | 0.092 | 0.125 | 0.094 | 0.145 | 0.148 | 0.069 | 0.096 | 0.020 |
| | KoVAE | 0.057 | 0.054 | 0.045 | 0.050 | 0.057 | **0.047** | 0.050 | 0.074 | 0.017 |
| | Ours | **0.053** | **0.046** | **0.044** | **0.044** | **0.022** | 0.049 | **0.047** | **0.069** | **0.012** |
| **FID** | TimeGAN-$\Delta t$ | 3.140 | 3.199 | 3.419 | 3.218 | 2.378 | 23.39 | 6.507 | 2.780 | 2.668 |
| | GT-GAN | 2.212 | 8.635 | 14.29 | 6.385 | 2.758 | 9.993 | 1.531 | 1.698 | 2.181 |
| | KoVAE | 1.518 | 0.248 | 0.180 | 0.280 | 3.699 | 6.163 | 0.629 | 0.037 | 0.369 |
| | Ours | **0.124** | **0.035** | **0.047** | **0.024** | **0.170** | **3.580** | **0.132** | **0.015** | **0.036** |
| **Corr.** | TimeGAN-$\Delta t$ | 3.743 | 1.051 | 2.350 | 0.579 | 1.200 | 13.24 | 3.765 | 2.424 | 1.399 |
| | GT-GAN | 7.148 | 0.916 | 2.467 | 0.356 | 0.791 | 14.92 | 3.889 | 3.282 | 0.261 |
| | KoVAE | 0.183 | 0.177 | 0.130 | 0.262 | 2.899 | 4.283 | 2.630 | 0.041 | 0.064 |
| | Ours | **0.084** | **0.054** | **0.065** | **0.039** | **0.396** | **2.031** | **0.922** | **0.015** | **0.019** |

Table 3: Averaged results over 30%, 50%, 70% missing rates for length 96 (top) and 768 (bottom).

| | | Model | ETTh1 | ETTh2 | ETTm1 | ETTm2 | Weather | Energy | Sine | Stock |
|---|---|---|---|---|---|---|---|---|---|---|
| **Length = 96** | **Disc.** | KoVAE | 0.284 | 0.103 | 0.276 | 0.089 | 0.341 | 0.356 | 0.239 | 0.099 |
| | | Ours | **0.070** | **0.053** | **0.040** | **0.030** | **0.152** | **0.185** | **0.003** | **0.017** |
| | **Pred.** | KoVAE | 0.062 | 0.057 | 0.054 | 0.050 | 0.046 | 0.080 | 0.165 | 0.020 |
| | | Ours | **0.053** | **0.049** | **0.045** | **0.044** | **0.025** | **0.049** | **0.155** | **0.011** |
| | **FID** | KoVAE | 5.842 | 1.111 | 4.070 | 0.996 | 3.681 | 4.694 | 4.381 | 0.868 |
| | | Ours | **0.357** | **0.508** | **0.171** | **0.142** | **0.394** | **0.309** | **0.017** | **0.109** |
| | **Corr.** | KoVAE | 0.224 | 0.362 | 0.175 | 0.435 | 2.609 | 4.810 | 0.222 | 0.089 |
| | | Ours | **0.114** | **0.151** | **0.092** | **0.094** | **0.890** | **1.161** | **0.016** | **0.014** |
| **Length = 768** | **Disc.** | KoVAE | 0.238 | 0.201 | 0.236 | 0.196 | 0.428 | 0.384 | 0.350 | 0.284 |
| | | Ours | **0.088** | **0.045** | **0.058** | **0.052** | **0.102** | **0.213** | **0.006** | **0.022** |
| | **Pred.** | KoVAE | 0.072 | 0.069 | 0.060 | 0.076 | 0.070 | 0.087 | 0.226 | 0.031 |
| | | Ours | **0.053** | **0.056** | **0.047** | **0.050** | **0.027** | **0.042** | **0.204** | **0.013** |
| | **FID** | KoVAE | 13.92 | 8.304 | 13.50 | 8.279 | 17.49 | 24.51 | 38.60 | 7.273 |
| | | Ours | **0.882** | **0.439** | **0.391** | **0.264** | **0.318** | **0.796** | **0.237** | **0.160** |
| | **Corr.** | KoVAE | 0.333 | 0.606 | 0.404 | 0.738 | 3.252 | 8.752 | 0.379 | 0.046 |
| | | Ours | **0.103** | **0.106** | **0.122** | **0.128** | **0.458** | **1.040** | **0.006** | **0.026** |

## 5.4 Irregularly-sampled data under noise

Our work primarily addresses irregularly sampled time series data. However, in real-world scenarios, such data often includes noise due to sensor limitations and inaccuracies. To further enhance our quantitative evaluation framework, we tackle the generative challenges of learning from irregular time series in noisy environments. Specifically, we propose a novel setup to evaluate the model's capability to recover the true underlying distribution from data corrupted by both irregular sampling and Gaussian noise. In this setup, we simulate a 50% missing rate and inject additive Gaussian noise sampled from a normal distribution $\mathcal{N}(0, \sigma)$, where $\sigma$ corresponds to the specified noise level (e.g., $0.1, 0.15,$ and $0.2$). Importantly, this noise is added independently of the original data distribution or scale. The evaluation is conducted on sequences of length 24 across four datasets: Weather, Etth1, Stocks. and Energy. Following the discriminative and predictive evaluation protocols for

Table 5: Discriminative and predictive scores for 50% missing rate on Weather, ETTh1, Stock, and Energy datasets with injected noise levels $(0.1, 0.15,$ and $0.2)$.

| N/R | Model | Weather Disc. | Pred. | ETTh1 Disc. | Pred. | Stock Disc. | Pred. | Energy Disc. | Pred. |
|---|---|---|---|---|---|---|---|---|---|
| 0.1 | KoVAE | 0.426 | 0.056 | 0.225 | 0.073 | 0.235 | 0.016 | 0.434 | 0.067 |
| | Ours | **0.061** | **0.052** | **0.024** | **0.034** | **0.007** | **0.012** | **0.065** | **0.047** |
| 0.15 | KoVAE | 0.488 | 0.092 | **0.377** | 0.077 | 0.341 | 0.092 | 0.493 | 0.093 |
| | Ours | **0.416** | **0.029** | 0.407 | **0.059** | **0.282** | **0.023** | **0.467** | **0.053** |
| 0.2 | KoVAE | 0.491 | 0.096 | **0.440** | 0.084 | 0.352 | 0.121 | 0.496 | 0.123 |
| | Ours | **0.485** | **0.035** | 0.456 | **0.062** | **0.340** | **0.027** | **0.457** | **0.057** |

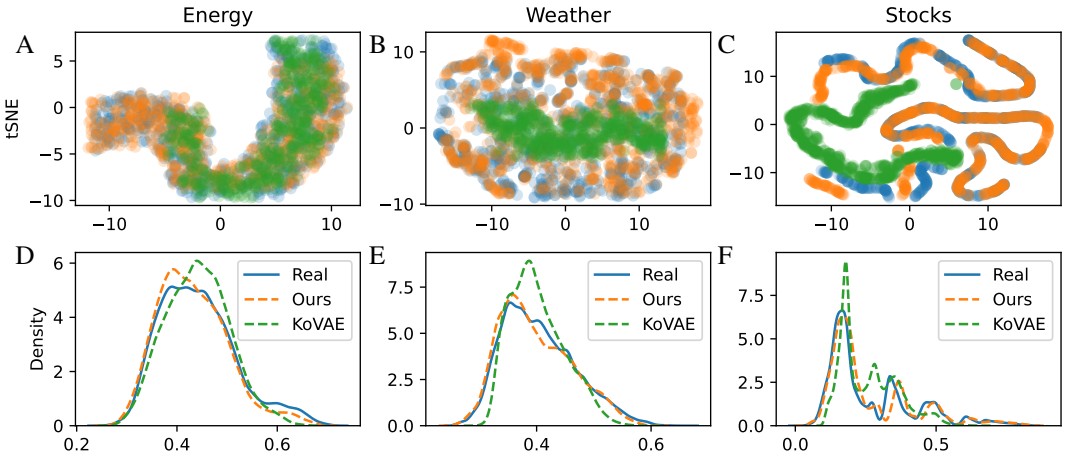

Figure 4: 2D t-SNE embeddings (top) and probability density functions (bottom) for real data vs. synthetic data from our method and KoVAE, under a 70% missing rate. From left to right: Energy (length 24), Weather (length 96), and Stock (length 768) datasets.

Table 6: Discriminative and predictive scores for 40% random vs. 40% continuous missingness on the Weather and Energy datasets with sequence lengths of 24 and 96. Lower is better.

| | Metric | Weather | | Energy | |
|---|---|---|---|---|---|
| | | Random | Continuous | Random | Continuous |
| Length = 24 | Discriminative ↓ | 0.057 | **0.053** | **0.081** | 0.082 |
| | Predictive ↓ | **0.021** | 0.025 | 0.047 | **0.043** |
| Length = 96 | Discriminative ↓ | **0.154** | 0.165 | **0.185** | 0.191 |
| | Predictive ↓ | 0.025 | **0.023** | **0.048** | 0.050 |

50% missing rate described earlier, we compare our approach against the most recent state-of-the-art method, KoVAE. Tab. 5 presents the results. For each noise rate (N/R), we report the discriminative (Disc.) and predictive (Pred.) scores, where lower values indicate better performance. Our method consistently outperforms KoVAE, achieving significant improvements. Specifically, we observe an average relative improvement of **26.8%** in the discriminative score and **50.3%** in the predictive score across all datasets and noise levels.

### 5.5 Robustness to missingness patterns

While many time series exhibit randomly missing values, in practice, missing values can also occur in contiguous blocks due to sensor outages or communication delays. To study this effect, we compare model performance under two types of 40% missingness: (i) randomly dropped values, and (ii) continuous missing blocks. This experiment is conducted on the Weather and Energy datasets using sequence lengths of 24 and 96. Discriminative and predictive scores are reported in Tab. 6. The results show that our method maintains strong performance under both missingness types, with some cases slightly favoring random missingness and others slightly favoring block-missingness. Overall, these findings confirm that our approach is robust across a range of missingness patterns, effectively handling both sporadic and structured data gaps.

### 5.6 Ablation Studies

To better understand the contributions of each component in our proposed architecture, we conducted a series of ablation studies. We explore each of the components in our approach separately, and, additionally, we modify recent approaches to include our completion strategy. Specifically, we consider the following models: (i) KoVAE + TST, where the NCDE module in KoVAE is replaced by TST, as in our approach. (ii) TimeAutoDiff [51] + TST (iii): TransFusion [48] + TST; The latter two baselines are diffusion-based models designed for regular time series but not specifically for irregular time series data. Therefore, we used the TST module to impute the missing values. (iv) Mask Only, where the TST autoencoder is removed, and we only apply the masking mechanism. In this setup, missing values are imputed using unnatural neighbors by filling them with zeros. (v) Ours Without

Table 7: Discriminative scores of the ablation study with 30%, 50%, and 70% drop-rate on Energy and Stock datasets for sequence lengths of $24, 96,$ and $768$.

| | Model | 30% | | 50% | | 70% | |
|---|---|---|---|---|---|---|---|
| | | Energy | Stock | Energy | Stock | Energy | Stock |
| **Len. = 24** | KoVAE + TST | 0.399 | 0.109 | 0.407 | 0.064 | 0.408 | 0.037 |
| | TimeAutoDiff + TST | 0.293 | 0.100 | 0.329 | 0.101 | 0.468 | 0.375 |
| | TransFusion + TST | 0.201 | 0.050 | 0.279 | 0.058 | 0.423 | 0.065 |
| | Ours (Mask Only) | 0.157 | 0.087 | 0.269 | 0.168 | 0.372 | 0.237 |
| | Ours (Without Mask) | 0.158 | 0.025 | 0.307 | 0.045 | 0.444 | 0.013 |
| | Ours | **0.048** | **0.007** | **0.065** | **0.007** | **0.128** | **0.007** |
| **Len. = 96** | KoVAE + TST | 0.240 | 0.185 | 0.254 | 0.221 | 0.417 | 0.193 |
| | TimeAutoDiff + TST | 0.299 | 0.105 | 0.336 | 0.104 | 0.461 | 0.398 |
| | TransFusion + TST | 0.305 | 0.083 | 0.335 | 0.098 | 0.442 | 0.116 |
| | Ours (Mask Only) | 0.490 | 0.174 | 0.422 | 0.263 | 0.480 | 0.388 |
| | Ours (Without Mask) | 0.402 | 0.033 | 0.476 | 0.072 | 0.491 | 0.082 |
| | Ours | **0.130** | **0.011** | **0.153** | **0.018** | **0.272** | **0.021** |
| **Len. = 768** | KoVAE + TST | 0.380 | 0.225 | 0.418 | 0.243 | 0.385 | 0.186 |
| | TimeAutoDiff + TST | 0.299 | 0.104 | 0.334 | 0.101 | 0.466 | 0.487 |
| | TransFusion + TST | 0.367 | 0.113 | 0.395 | 0.121 | 0.451 | 0.131 |
| | Ours (Mask Only) | 0.437 | 0.249 | 0.349 | 0.450 | 0.435 | 0.491 |
| | Ours (Without Mask) | 0.364 | 0.027 | 0.353 | 0.102 | 0.325 | 0.102 |
| | Ours | **0.170** | **0.025** | **0.244** | **0.033** | **0.251** | **0.013** |

Masking, where we leverage TST to complete missing values, and training is performed without masking. (vi) Our approach.

We quantitatively evaluated each model under the same experimental conditions and show the results in Tab. 7. To provide an extensive analysis, our tests include several missing rates $(30\%, 50\%, \text{and } 70\%)$ using two datasets, Energy and Stock. Further, we measured the performance across different sequence lengths $(24, 96, \text{and } 768)$. Our findings show that the combination of TST-based completion and masking yields superior performance compared to all other setups. Specifically, the Mask Only and Ours (Without Masking) setups showed significant limitations in capturing the true data distribution, while the replacement of NCDE with TST (KoVAE + TST) fell short in comparison to our proposed architecture. In particular, our results reveal that replacing the NCDE imputation component in KoVAE with the TST imputation mechanism is not the primary factor driving the significant improvements achieved by our method. Moreover, even when employing a powerful time series diffusion-based model like TransFusion combined with TST-based imputation, performance significantly degrades, struggling to capture the true distribution of the regular data. Overall, these results highlight the critical role of masking during the diffusion process and the importance of leveraging completion as a guide rather than a direct substitute for the true distribution.

We also ablate the impact of different image transformations on model performance, evaluating four methods: vanilla folding (reshapes a sequence into a fixed-size matrix with zero-padding), Gramian Angular Field, basic delay embedding, and our proposed inverse delay embedding (see App. A). Results in Tab. 8a and Tab. 9 show that geometric approaches—vanilla folding and our enhanced DE—better suit our method due to their structural clarity, where each pixel maps directly to a time point, facilitating mask usage and improving both discriminative and predictive performance. In contrast, GAF does not scale well to long sequences due to large image size. Our inverse transform also outperforms the original inverse of ImagenTime [38].

We additionally conducted an extensive ablation study comparing a variety of completion strategies. These included simple methods such as, Gaussian noise (GN), zero-filling, linear interpolation (LI), and polynomial interpolation (PI); probabilistic techniques like stochastic imputation (SI) (sampling from a Gaussian distribution fitted to the non-NaN values in each slice); and more advanced learning-based approaches, including NCDE, CSDI [53], and our proposed Time Series Transformer (TST) completion. Our results in Tab. 8b show that when the neighbors are not natural—such as in the case of zero completion or Gaussian noise completion—the model struggles more to generate data that closely follows the true distribution. In contrast, when using more natural completions (e.g., polynomial, stochastic imputation, NCDE, CSDI, TST), the model consistently obtains very good results. This confirms that generating natural neighborhoods indeed enhances the generative quality without making the model completely reliant on the imputation quality.

Table 8: Comparative performance on Energy and Stock datasets for sequence length 24.

(a) Ablation study on image transformation methods.

| | Model | 30% | | 50% | | 70% | |
|---|---|---|---|---|---|---|---|
| | | Energy | Stock | Energy | Stock | Energy | Stock |
| **Disc.** | Gramian Angular | 0.291 | 0.061 | 0.313 | 0.157 | 0.363 | 0.183 |
| | Vanilla Folding | 0.058 | **0.005** | **0.050** | 0.009 | 0.136 | 0.010 |
| | Basic DE | 0.091 | 0.035 | 0.102 | 0.046 | 0.153 | 0.019 |
| | Ours DE | **0.048** | 0.007 | 0.065 | **0.007** | **0.128** | **0.007** |
| **Pred.** | Gramian Angular | 0.049 | 0.013 | 0.048 | 0.015 | 0.049 | 0.016 |
| | Vanilla Folding | 0.047 | 0.013 | 0.047 | 0.013 | 0.047 | 0.014 |
| | Basic DE | 0.053 | 0.022 | 0.051 | 0.025 | 0.055 | 0.027 |
| | Ours DE | **0.047** | **0.012** | **0.047** | **0.012** | **0.047** | **0.011** |

(b) Imputation methods with 50% drop-rate.

| Model | Disc. | | Pred. | |
|---|---|---|---|---|
| | Energy | Stock | Energy | Stock |
| $GN \rightarrow$ NaN | 0.457 | 0.102 | 0.058 | 0.014 |
| $0 \rightarrow$ NaN | 0.269 | 0.158 | 0.051 | 0.014 |
| LI | 0.251 | 0.013 | 0.049 | 0.019 |
| PI | 0.201 | 0.012 | 0.053 | 0.016 |
| NCDE | 0.102 | 0.013 | 0.058 | 0.013 |
| CSDI | 0.088 | 0.012 | 0.048 | 0.013 |
| SI | 0.069 | 0.010 | 0.047 | 0.013 |
| Ours (TST) | **0.065** | **0.007** | **0.047** | **0.012** |

## 6 Conclusions

In this work, we introduced a novel two-step framework for generating realistic regular time series from irregularly sampled sequences. By integrating a Time Series Transformer (TST) for completion with a vision-based diffusion model leveraging masking, we effectively addressed the challenge of unnatural neighborhoods inherent in direct masking approaches. This hybrid strategy ensures that the diffusion model benefits from more structured and meaningful input while mitigating over-reliance on completed values. Our extensive evaluations across multiple benchmarks demonstrated state-of-the-art performance, with improvements of up to 70% in discriminative score and an 85% reduction in computational cost over prior methods. Furthermore, our approach scales effectively to long time series, significantly outperforming existing generative models in both accuracy and efficiency.

Beyond these advancements, our work highlights the broader potential of integrating completion and masking strategies in generative modeling, particularly in domains where irregular sampling and missing values are prevalent. Future directions include extending our framework to multimodal time series generation, exploring self-supervised objectives for improved imputation, and integrating adaptive masking techniques that dynamically adjust completion reliance. By bridging the gap between irregular and regular time series generation, our method opens new possibilities for high-fidelity synthetic data generation in critical fields such as healthcare, finance, and climate science.

## Acknowledgments

This research was partially supported by the Lynn and William Frankel Center of the Computer Science Department, Ben-Gurion University of the Negev, an ISF grant 668/21, an ISF equipment grant, and by the Israeli Council for Higher Education (CHE) via the Data Science Research Center, Ben-Gurion University of the Negev, Israel.

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

# A  Time Series-to-Image Transformation

In this section, we provide a brief discussion on time series to image conversions. We then introduce Delay Embedding, the transformation we employed. Additionally, we describe an improvement we implemented to enhance the reversibility of delay embedding.

**Time series to image conversion.**   The conversion of time series data into image representations has attracted significant interest for its ability to leverage computer vision methods in time series analysis. Techniques like Gramian Angular Fields [57], Recurrence Plots [19], and Line Graphs [32] allow for mapping time series data to visual formats, enabling tasks such as classification and imputation. In the field of speech analysis, the short-time Fourier transform (STFT) [1, 2, 56, 15] is crucial for capturing frequency variations over time, which is vital for processing audio and speech data. Recent advancements have also explored incorporating mel-spectrograms in diffusion models, particularly in connection with latent diffusion spaces[42, 10, 36]. Furthermore, generative approaches such as Wasserstein GANs [7, 20] have been applied to time series in image form.

**Vanilla Folding**   is a straightforward transformation. Given a time series $x$, we reshape it into an image $x_{\text{img}}$ by filling rows from left to right and moving to the next row upon reaching the end, padding with zeros if necessary. The inverse transformation reconstructs the original time series by reading the non-padded region row-wise. Despite its simplicity, this method scales well to very long sequences. Folding can also be interpreted as a specific case of delay embedding, as we explain below.

**Delay Embedding and Enhanced Reverse Transformation**   [52] converts a univariate time series $x_{1:L} \in \mathbb{R}^L$ into an image by structuring its information into columns and applying zero-padding as needed. This transformation is controlled by two hyperparameters: $m$, which defines the skip value, and $n$, which determines the number of columns. Given any channel of a time series, the corresponding matrix $X$ is formed as follows:

$$
X = \begin{bmatrix} x_1 & x_{m+1} & \dots & x_{L-n} \\ \vdots & \vdots & \dots & \vdots \\ x_n & x_{n+m+1} & \dots & x_L \end{bmatrix} \in \mathbb{R}^{n \times q} \, ,
$$

where $q = \lceil (L-n)/m \rceil$. To match the input requirements of a neural network, the resulting image $x_{\text{img}}$ is padded with zeros. This transformation is applied independently to each channel, and for multivariate time series, the matrices $X$ from different channels are concatenated along a new axis. Given an input signal $x \in \mathbb{R}^{L \times K}$, the output is a transformed representation $x_{\text{img}} \in \mathbb{R}^{K \times n \times q}$, which is then zero-padded to obtain $x_{\text{img}} \in \mathbb{R}^{K \times n \times n}$. Delay embedding efficiently scales to long sequences; for instance, choosing $m = n = 256$ enables the encoding of sequences up to $65k$ in length using $256 \times 256$ images.

Our primary innovation lies in the reverse transformation process. In the original approach, only the first pixel corresponding to each time series value is used for reconstruction. Specifically, if $x_i$ is mapped to multiple image indices, the original method selects the first corresponding pixel in the image for reconstruction. In contrast, our approach aggregates information from all corresponding image indices by computing the average of the associated pixels for each $x_i$. For a given $x_{1:L} \in \mathbb{R}^L$, both methods ensure that $f^{-1}(f(x)) = x$.

As shown in Table 9, our approach consistently outperforms the original approach across various drop rates. For example, at a 30% drop rate, our delay embedding (DE) approach achieves a discriminative score of 0.020 for the ETTh1 dataset and 0.009 for ETTh2, compared to 0.023 and 0.018 with the naive approach, respectively. Similarly, at a 50% drop rate, the new DE method reduces the discriminative score to 0.032 (ETTh1) and 0.005 (ETTh2), outperforming the previous DE approach, which yielded scores of 0.040 and 0.037, respectively.

Table 9: Discriminative scores on ETTh1 and ETTh2 for sequence length 24 to compare the original inverse delay embedding vs. our inverse, evaluated in $30\%, 50\%,$ and $70\%$ missing rates.

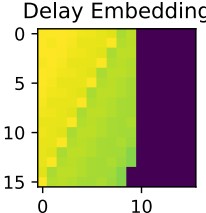

Delay Embedding

| Model | 30% | | 50% | | 70% | |
|---|---|---|---|---|---|---|
| | ETTh1 | ETTh2 | ETTh2 | ETTh2 | ETTh2 | ETTh2 |
| Ours + Old inverse | 0.023 | 0.018 | 0.040 | 0.037 | 0.067 | 0.048 |
| Ours + New inverse | **0.020** | **0.009** | **0.032** | **0.005** | **0.058** | **0.013** |

## B    Training Losses

Our model consists of two main components: an Autoencoder (AE) and a vision diffusion model (ImagenTime). For each component, we modified the loss to handle irregular data more effectively.

### B.1    Autoencoder Training Loss

The TST-Based AE is trained to reconstruct only the known (non-missing) values of the input data. Since we do not have access to the regularly sampled time series during training, the model learns to infer missing values from the irregularly sampled data. The masked reconstruction loss is defined as:

$$\mathcal{L}_e^{t=0} = \frac{1}{|\mathcal{O}|} \sum_{i \in \mathcal{O}} (\tilde{x}_i - x_i)^2 \tag{1}$$

where $x$ is the original input data, $\tilde{x}$ is the reconstructed output, $\mathcal{O}$ represents the set of observed (non-missing) indices in the input $x$, and $|\mathcal{O}|$ is the number of observed values. This ensures that the loss function only penalizes reconstruction errors for the known values, without considering the missing ones.

### B.2    ImagenTime Diffusion Training Loss

At the core of **ImagenTime** is the generative diffusion model, which follows the framework of Karras et al. [27] for improved score-based modeling. The model employs a second-order ODE for the reverse process, balancing fast sampling and high-quality generations.

For irregular time series data, we changed the training loss to ensure proper weighting of the diffusion steps and account for missing values. Given an input sequence $x$ and a corresponding mask $m$ indicating observed entries, the loss is defined as:

$$\mathcal{L}_{\text{diff}} = \mathbb{E}_{x,\sigma} \left[ \|(D_\theta(x + \sigma n, \sigma) - x) \cdot m\|^2 \right] \tag{2}$$

where:

- $x$ is the input time series with missing values.
- $n \sim \mathcal{N}(0, I)$ is standard Gaussian noise, scaled by $\sigma$.
- $m$ is the mask, indicating the observed values.

The model reconstructs the output using both observed and imputed indices, treating the imputed values as natural neighbors to the observed ones. However, the loss is computed and compared only against the observed indices, ensuring that the comparison is made solely to the true distribution, unaffected by the imputed values. This allows the model to learn the distribution while maintaining accurate reconstruction of missing values.

## C    Inference Time Analysis

In this section, we present a detailed comparison of the inference time between our approach and KoVAE, particularly in relation to sequence length. The evaluation is conducted across a range of sequence lengths: 24, 96, and 768. Additionally, we explore the relationship between inference time and sequence length for both models.

## C.1 Inference Time vs. Sequence Length

Figure 5 illustrates the relationship between inference time per sample (in seconds) and sequence length for both our model and KoVAE. While KoVAE demonstrates faster sampling on short sequences, its efficiency degrades rapidly as the sequence length increases. KoVAE, which is based on a sequential VAE architecture for time series, processes each time step individually throughout the entire sequence, causing its computational cost to increase significantly with longer sequences and resulting in substantially longer inference times.

In contrast, our model maintains nearly constant inference time regardless of sequence length, making it highly efficient even for sequences as long as 5000 time steps. This is due to the use of delay embedding, which enables the model to compress long sequences into compact image representations with fixed dimensions. A clear turning point occurs at a sequence length of approximately 4500, beyond which our model consistently outperforms KoVAE in terms of inference time. This robust performance highlights the advantage of our method in terms of time efficiency, especially when dealing with long sequences.

## C.2 Inference Time and Fidelity Comparison

As shown in Figure 6, we evaluate the relationship between inference time and fidelity (indicated by the discriminative score) for a single sequence. Lower discriminative scores correspond to higher fidelity in the model's predictions. On the other hand, inference time is a measure of efficiency, with shorter times indicating greater computational efficiency.

Our approach consistently performs well by maintaining a low discriminative score, which translates into higher fidelity in its predictions. This is achieved while also managing to keep the inference time relatively low, even for longer sequences. Notably, our model uses only 18 sampling steps regardless of sequence length, which results in stable and fast inference times across all configurations.

## C.3 Performance Comparison

As observed from both figures, our model exhibits a favorable trade-off between inference efficiency and fidelity. While KoVAE might be more efficient for shorter sequences, its performance degrades as sequence length increases, making it less suitable for long sequences. Our approach, however, remains consistently efficient and accurate, maintaining both low discriminative scores and linear inference times as sequence lengths scale.

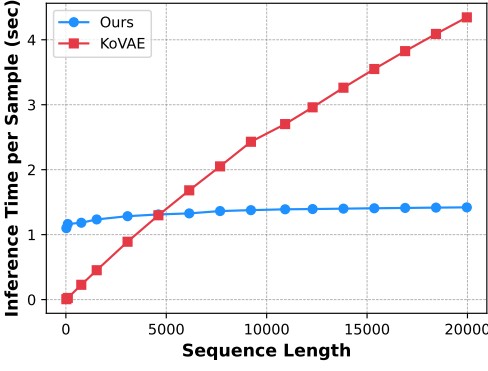

Figure 5: Comparison of inference time per sample in seconds vs. sequence length of our model and KoVae model.

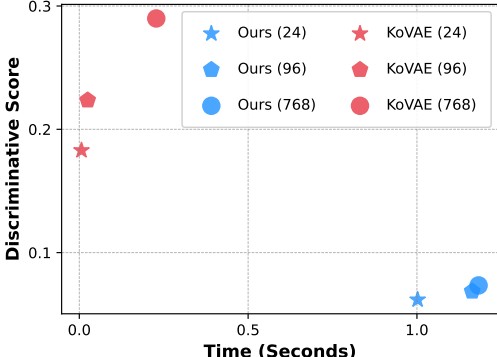

Figure 6: Comparison of discriminative score vs. inference time of a single sequence for our approach and KoVAE across different sequence lengths (24, 96, and 768). Lower discriminative scores indicate higher fidelity, and shorter inference times reflect greater efficiency.

# D Experimental Setup

## D.1 Baseline Methods

We compare our method with several generative time series models designed for irregular data. KoVAE [39] is specifically designed to handle irregularly sampled time series effectively. Additionally, we consider GT-GAN [26], another method tailored for irregular time series generation. Lastly, we evaluate against TimeGAN-$\Delta t$ [58], a re-designed version of the original TimeGAN. Since TimeGAN does not natively support irregular time series, we follow GT-GAN and compare them with their re-designed versions. Specifically, extending regular approaches to support irregular TS requires the conversion of a dynamical module to its time-continuous version. we adapted it by converting its GRU layers to GRU-$\Delta t$, enabling the model to exploit the time differences between observations and capture temporal dynamics.

## D.2 Datasets

We conduct experiments using a combination of synthetic and real-world datasets, each designed to evaluate the model under various conditions, including regular and irregular time-series settings.

**Sines** This synthetic dataset contains 5 features, where each feature is independently generated using sinusoidal functions with different frequencies and phases. Specifically, for each feature $i \in \{1, ..., 5\}$, the time-series data is defined as $x_i(t) = \sin(2\pi f_i t + \theta_i)$, where $f_i \sim U[0, 1]$ and $\theta_i \sim U[-\pi, \pi]$. The dataset is characterized by its continuity and periodic properties, making it a suitable benchmark for evaluating the model's ability to handle structured time-series data.

**Stocks** The Stocks dataset comprises daily historical Google stock price data from 2004 to 2019. It includes six features: high, low, opening, closing, adjusted closing prices, and trading volume. Unlike Sines, this dataset lacks periodicity and primarily exhibits random walk patterns. It is a real-world dataset commonly used to benchmark financial time-series forecasting and modeling.

**MuJoCo** MuJoCo (Multi-Joint dynamics with Contact) is a versatile physics simulation framework used to generate multivariate time-series data [54]. The dataset contains 14 features representing state variables and control actions from simulated trajectories. This dataset is particularly suitable for evaluating models on dynamical systems and tasks involving physical interactions

**Energy** The Energy dataset is a real-world multivariate time-series dataset [8] derived from the UCI Appliance Energy Prediction dataset. It includes 28 features, which are correlated and exhibit noisy periodicity and continuous-valued measurements. This dataset provides a challenging benchmark for forecasting and modeling tasks involving environmental and appliance energy consumption data.

**ETTh & ETTm** The ETTh (Electricity Transformer Temperature - Hourly) and ETTm (Electricity Transformer Temperature - Minute) datasets [62] capture electricity load data from two power stations with varying temporal resolutions. These datasets are used for short- and long-term time-series forecasting tasks and are part of an established benchmark for evaluating generative and predictive models.

**Weather** The Weather dataset includes daily meteorological measurements, such as temperature, precipitation, snowfall, snow depth, and minimum and maximum temperatures, collected from the United States Historical Climatology Network (USHCN) [2] . The dataset comprises measurements from 1,218 weather stations and is used for analyzing climatic trends and weather forecasting tasks.

**Electricity** The Electricity dataset consists of electricity consumption data across multiple clients, represented as multivariate time-series. It is widely used for forecasting electricity loads and understanding temporal consumption patterns in energy-related applications.

---

[2] https://knb.ecoinformatics.org/view/doi%3A10.3334%2FCDIAC%2FCLI.NDP019

**Traffic** The Traffic dataset [23] contains hourly traffic volume data for westbound I-94 in the Minneapolis-St. Paul, MN area, collected from 2012 to 2018. It includes eight features, mixing numerical measurements (e.g., temperature, rainfall, snowfall, cloud coverage, traffic volume) with several categorical variables (e.g., holiday indicators, weather descriptions), making it one of the few benchmarks that requires models to handle both continuous and categorical time-series data. The dataset captures multivariate, sequential patterns influenced by weather and holiday effects, making it particularly suitable as a generative benchmark for modeling complex dependencies across heterogeneous features.

## D.3 Metrics

**Discriminative Score** This metric measures the ability of a model to differentiate between real and generated data. A lower discriminative score indicates that the generated data is more indistinguishable from the real data, reflecting better generative performance. This score is typically computed by training a binary classifier and evaluating its accuracy in distinguishing between the two datasets. A score close to random guessing suggests that the synthetic data is nearly indistinguishable from real data.

**Predictive Score** The predictive score evaluates the quality of the generated data in terms of its utility for downstream predictive tasks. It is typically assessed using a supervised learning model trained on generated data and tested on real data, or vice versa. A higher predictive score indicates better alignment between real and generated distributions.

**Context-FID Score** A lower Frechet Inception Distance (FID) score indicates that synthetic sequences are more similar to the original data distribution. Paul et al. (2022) introduced a variation of FID, called Context-FID (Context-Frechet Inception Distance), which replaces the Inception model in the original FID calculation with TS2Vec, a time series representation learning method [60]. Their findings suggest that models with the lowest Context-FID scores tend to achieve the best results in downstream tasks. Additionally, they demonstrated a strong correlation between the Context-FID score and the forecasting performance of generative models. To compute this score, synthetic and real time series samples are first generated, then encoded using a pre-trained TS2Vec model, after which the FID score is calculated based on the learned representations.

**Correlational Score** Building on the approach from [33], we estimate the covariance between the $i^{th}$ and $j^{th}$ features of a time series using the following formula:

$$\text{cov}_{i,j} = \frac{1}{T} \sum_{t=1}^{T} X_t^i X_t^j - \left( \frac{1}{T} \sum_{t=1}^{T} X_t^i \right) \left( \frac{1}{T} \sum_{t=1}^{T} X_t^j \right).$$

To quantify the correlation between real and synthetic data, we compute the following metric:

$$\frac{1}{10} \sum_{i,j} \left| \frac{\text{cov}_r^{i,j}}{\sqrt{\text{cov}_r^{i,i} \text{cov}_r^{j,j}}} - \frac{\text{cov}_f^{i,j}}{\sqrt{\text{cov}_f^{i,i} \text{cov}_f^{j,j}}} \right|$$

## D.4 Hyperparameters.

We summarize the key hyperparameters used in our framework in Tables 10, 11, and 12, corresponding to sequence lengths of 24, 96, and 768, respectively. The hyperparameters remain largely consistent across tasks, with variations in batch size, embedding dimensions, and image resolutions. We use the default EDM [27] sampler for all datasets and follow a unified configuration for the U-Net architecture in the diffusion model. For further details, refer to [27]. Additionally, all models were trained using the same learning rate schedule and optimization settings to ensure comparability across different sequence lengths.

Table 10: Hyperparameter Settings for Sequence Length 24 Across Different Datasets

| | ETTh1 | ETTh2 | ETTm1 | ETTm2 | Weather | Electricity | Energy | Sine | Stock | Mujoco |
|---|---|---|---|---|---|---|---|---|---|---|
| **General** | | | | | | | | | | |
| *image size* | $16 \times 16$ | $16 \times 16$ | $16 \times 16$ | $16 \times 8$ | $8 \times 8$ | $8 \times 8$ | $8 \times 8$ | $8 \times 8$ | $8 \times 8$ | $8 \times 8$ |
| *learning rate* | $10^{-4}$ | $10^{-4}$ | $10^{-4}$ | $10^{-4}$ | $10^{-4}$ | $10^{-4}$ | $10^{-4}$ | $10^{-4}$ | $10^{-4}$ | $10^{-4}$ |
| *batch size* | 128 | 128 | 128 | 128 | 128 | 128 | 128 | 128 | 128 | 128 |
| *teacher forcing rate* | 0 | 0 | 0 | 0 | 0 | 0 | 0 | 0 | 0.2 | 0 |
| **DE** | | | | | | | | | | |
| *embedding(n)* | 8 | 8 | 8 | 8 | 8 | 8 | 8 | 8 | 8 | 8 |
| *delay(m)* | 3 | 3 | 3 | 3 | 3 | 3 | 3 | 3 | 3 | 3 |
| **TST** | | | | | | | | | | |
| *hidden_dim* | 40 | 40 | 40 | 40 | 40 | 40 | 40 | 40 | 40 | 40 |
| *n_heads* | 5 | 5 | 5 | 5 | 5 | 5 | 5 | 5 | 5 | 5 |
| *num_layers* | 6 | 6 | 6 | 6 | 6 | 6 | 6 | 6 | 6 | 6 |
| **Diffusion** | | | | | | | | | | |
| *U-net channels* | 128 | 128 | 128 | 128 | 128 | 128 | 128 | 128 | 128 | 64 |
| *in channels* | $[1,2,2,2]$ | $[1,2,2,2]$ | $[1,2,2,2]$ | $[1,2,2,2]$ | $[1,2,2,4]$ | $[1,2,2,4]$ | $[1,2,2,4]$ | $[1,2,2,2]$ | $[1,2,2,2]$ | $[1,2,2,2]$ |
| *attention revolution* | $[8,4,2]$ | $[8,4,2]$ | $[8,4,2]$ | $[8,4,2]$ | $[8,4,2]$ | $[8,4,2]$ | $[8,4,2]$ | $[8,4,2]$ | $[8,4,2]$ | $[8,4,2]$ |
| *sampling steps* | 18 | 18 | 18 | 18 | 18 | 18 | 18 | 18 | 18 | 18 |

Table 11: Hyperparameter Settings for Sequence Length 96 Across Different Datasets

| | ETTh1 | ETTh2 | ETTm1 | ETTm2 | Weather | Energy | Sine | Stock |
|---|---|---|---|---|---|---|---|---|
| **General** | | | | | | | | |
| *image size* | $16 \times 16$ | $16 \times 16$ | $16 \times 16$ | $16 \times 16$ | $32 \times 32$ | $32 \times 32$ | $16 \times 16$ | $16 \times 16$ |
| *learning rate* | $10^{-4}$ | $10^{-4}$ | $10^{-4}$ | $10^{-4}$ | $10^{-4}$ | $10^{-4}$ | $10^{-4}$ | $10^{-4}$ |
| *batch size* | 32 | 64 | 128 | 128 | 32 | 128 | 128 | 16 |
| *teacher forcing rate* | 0 | 0 | 0 | 0 | 0 | 0 | 0 | 0.2 |
| **DE** | | | | | | | | |
| *embedding(n)* | 16 | 16 | 16 | 16 | 32 | 32 | 16 | 16 |
| *delay(m)* | 6 | 6 | 6 | 6 | 24 | 24 | 6 | 6 |
| **TST** | | | | | | | | |
| *hidden_dim* | 40 | 40 | 40 | 40 | 40 | 40 | 40 | 40 |
| *n_heads* | 5 | 5 | 5 | 5 | 5 | 5 | 5 | 5 |
| *num_layers* | 6 | 6 | 6 | 6 | 6 | 6 | 6 | 6 |
| **Diffusion** | | | | | | | | |
| *U-net channels* | 128 | 128 | 128 | 128 | 128 | 128 | 128 | 128 |
| *in channels* | $[1,2,2,2]$ | $[1,2,2,2]$ | $[1,2,2,2]$ | $[1,2,2,2]$ | $[1,2,2,4]$ | $[1,2,2,4]$ | $[1,2,2,2]$ | $[1,2,2,2]$ |
| *attention revolution* | $[16,8,4,2]$ | $[16,8,4,2]$ | $[16,8,4,2]$ | $[16,8,4,2]$ | $[32,16,8,4,2]$ | $[32,16,8,4,2]$ | $[16,8,4,2]$ | $[16,8,4,2]$ |
| *sampling steps* | 18 | 18 | 18 | 18 | 18 | 18 | 18 | 18 |

Table 12: Hyperparameter Settings for Sequence Length 768 Across Different Datasets

| | ETTh1 | ETTh2 | ETTm1 | ETTm2 | Weather | Energy | Sine | Stock |
|---|---|---|---|---|---|---|---|---|
| **General** | | | | | | | | |
| *image size* | $32 \times 32$ | $32 \times 32$ | $32 \times 32$ | $32 \times 32$ | $32 \times 32$ | $32 \times 32$ | $32 \times 32$ | $32 \times 32$ |
| *learning rate* | $10^{-4}$ | $10^{-4}$ | $10^{-4}$ | $10^{-4}$ | $10^{-4}$ | $10^{-4}$ | $10^{-4}$ | $10^{-4}$ |
| *batch size* | 32 | 32 | 32 | 32 | 32 | 16 | 32 | 32 |
| *teacher forcing rate* | 0 | 0 | 0 | 0 | 0 | 0 | 0 | 0.2 |
| **DE** | | | | | | | | |
| *embedding(n)* | 32 | 32 | 32 | 32 | 32 | 32 | 32 | 32 |
| *delay(m)* | 24 | 24 | 24 | 24 | 24 | 24 | 24 | 24 |
| **TST** | | | | | | | | |
| *hidden_dim* | 40 | 40 | 40 | 40 | 40 | 40 | 40 | 40 |
| *n_heads* | 5 | 5 | 5 | 5 | 5 | 5 | 5 | 5 |
| *num_layers* | 6 | 6 | 6 | 6 | 6 | 6 | 6 | 6 |
| **Diffusion** | | | | | | | | |
| *U-net channels* | 128 | 128 | 128 | 128 | 128 | 128 | 128 | 128 |
| *in channels* | $[1,2,2,2]$ | $[1,2,2,2]$ | $[1,2,2,2]$ | $[1,2,2,2]$ | $[1,2,2,4]$ | $[1,2,2,4]$ | $[1,2,2,2]$ | $[1,2,2,2]$ |
| *attention revolution* | $[32,16,8,4,2]$ | $[32,16,8,4,2]$ | $[32,16,8,4,2]$ | $[32,16,8,4,2]$ | $[32,16,8,4,2]$ | $[32,16,8,4,2]$ | $[32,16,8,4,2]$ | $[32,16,8,4,2]$ |
| *sampling steps* | 18 | 18 | 18 | 18 | 18 | 18 | 18 | 18 |

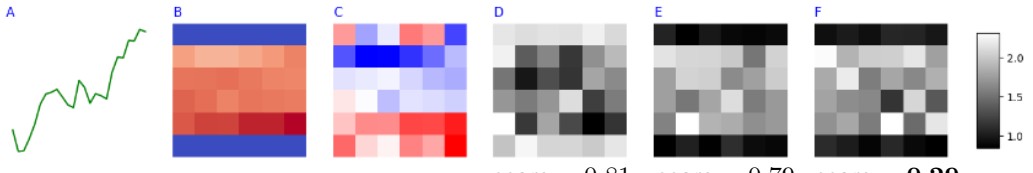

score = 0.81    score = 0.79    score = **0.29**

Figure 7: Unnatural vs. natural neighborhoods on stock data. A data point (A) is mapped to an image with zeros and its coordinates centered (B). Denoising the entire image produces inferior kernels (D) compared to masking (E). Constructing natural neighborhoods (C) yields consistent kernels and improved scores (F).

## D.5  Natural vs. Unnatural Neighborhoods Experiment Setup

In this section, we provide a detailed explanation of the experimental setup introduced in Sec. 4.

**Experiment Setup.**  We first generate $1000$ two-dimensional samples $\{p\}$ drawn from a multivariate Gaussian distribution, with means centered at $(1, 1)$, $(1, -1)$, $(-1, 1)$, and $(-1, -1)$. A figure illustrating all sampled data appears in Figure 1A. To better simulate our real world environment, we transform each 2D datapoint into a $3 \times 4$ image by setting all pixels to zero except those at the center, which correspond to the $x$ and $y$ coordinates of the original point (e.g., $s[1, 1] = p[0]$ and $s[1, 2] = p[1]$). Figure 1B depicts an example of this transformation. We refer to this dataset as $\mathcal{S}_{\text{irregular}}$, as it simulates an "irregular" dataset containing zeros for missing values.

Next, we construct a second dataset, $\mathcal{S}_{\text{regular}}$, in which the zero entries are replaced with linear or nonlinear transformations of $p[0]$, $p[1]$, or both. This is intended to emulate a data-imputation step performed on the fist step of our method, yielding a more "natural" neighborhood of values. An data point example can be found in Figure 1C. We then compare three training setups:

1. Train a diffusion model to predict the score across the *entire* image (i.e., noise prediction in diffusion) on $\mathcal{S}_{\text{irregular}}$.

2. Train a diffusion model to predict *only* the two central (coordinate) pixels, using our masking technique, on $\mathcal{S}_{\text{irregular}}$.

3. Train the same masked model as in (2), but on $\mathcal{S}_{\text{regular}}$, where we have "natural neighbors."

All score models share a simple architecture: a `Conv2D` layer with a $3 \times 4$ kernel, followed by a `ReLU` activation, and a deconvolution layer that restores the original input size.

**Evaluation.**  We employ two metrics:

1. **Score Estimation Loss:** For fair comparison, we measure the score prediction error on the two central pixels (i.e., the original coordinates), regardless of the training strategy.

2. **Kernel Analysis:** We inspect the first-layer convolution kernels to determine which pixels the model focuses on. Since there are $64$ output channels, we compute the $L_1$ norm at each spatial position across all channels and then average them.

## E  Additional Experiments and Analysis

### E.1  "Unnatural" Neighborhoods: Stock Market Experiment

As described in the main paper, we extend the synthetic experiment to a real-world dataset of stock prices. Implementation details appear in the main text; here we present the visual results (moved to the appendix due to space constraints). See Fig. 7.

### E.2  Extending to Categorical and Numerical Features

We further extend our evaluation to demonstrate that our method can handle mixed-type time series containing both numerical and categorical features. Categorical variables are mapped to learnable

Table 13: Discriminative scores on mixed-type data (Traffic) with 30%, 50%, and 70% drop-rate for sequence lengths of 24, and 96.

|  | Model | 30% | 50% | 70% |
|---|---|---|---|---|
| Len. **= 24** | GT-GAN | 0.481 | 0.473 | 0.485 |
|  | KoVAE | 0.154 | 0.172 | 0.222 |
|  | Ours | **0.061** | **0.064** | **0.087** |
| Len. **= 96** | GT-GAN | 0.493 | 0.491 | 0.488 |
|  | KoVAE | 0.212 | 0.245 | 0.307 |
|  | Ours | **0.073** | **0.091** | **0.102** |

embedding vectors, which are jointly optimized with the generative model, transforming the input into a fully continuous representation that the diffusion process can operate on seamlessly. During generation, embeddings are mapped back to discrete categories using a simple postprocessing step. To validate this capability, we evaluated our method on the Traffic dataset which includes both numerical and categorical attributes, across multiple sequence lengths and missing rates. As reported in Tab. 13, our approach achieves the best discriminative scores compared to prior methods, demonstrating its ability to effectively capture temporal dependencies in mixed-type time series while maintaining superior generative performance.

### E.3 Computational Efficiency of Completion Strategies

In addition to performance, we also compare the computational efficiency of completion strategies. Both NCDE and CSDI rely on costly operations—NCDE requires repeated cubic spline evaluations, while CSDI involves up to 1000 sampling steps per imputation—which makes them slow or infeasible for long sequences or high-dimensional data. In contrast, TST is lightweight and scales efficiently while still achieving competitive or superior results. As shown in Tab. 14, TST consistently trains faster and imputes orders of magnitude quicker than NCDE and CSDI; for instance, on the Energy dataset with sequence length 768, NCDE could not be trained due to memory limits and CSDI required over 84 hours of training and 2394 minutes for imputing 1024 samples, whereas TST completed training in just 6.67 hours and imputed all samples in only 6.26 seconds. These findings highlight TST's role as a scalable and efficient completion module, combining both high-quality generative performance and practical usability.

Table 14: Training (200 epochs) and imputation (1024 samples) times on RTX 3090. TST is significantly faster than NCDE and CSDI, especially for long sequences; NCDE times are omitted where training was infeasible.

| Dataset | Seq. Len | TST (Ours) | | NCDE | | CSDI | |
|---|---|---|---|---|---|---|---|
|  |  | Train (**h**) | Impute (**s**) | Train (**h**) | Impute (**s**) | Train (**h**) | Impute (**min**) |
| Energy | 24 | **0.67** | **0.64** | 2.55 | 2.4 | 1.60 | 35.71 |
|  | 96 | **0.80** | **0.74** | 7.68 | 5.6 | 5.43 | 135.29 |
|  | 768 | **6.67** | **6.26** | – | – | 84.61 | 2394.71 |
| Stock | 24 | **0.10** | **0.19** | 0.67 | 2.0 | 0.18 | 15.40 |
|  | 96 | **0.13** | **0.20** | 2.45 | 7.2 | 0.40 | 46.78 |
|  | 768 | **1.10** | **1.20** | 59.32 | 56.0 | 3.55 | 514.08 |

### E.4 Effect of Integrated Training Scheme

We investigate the impact of our integrated training scheme, which combines a short pre-training of the TST-based imputation module with joint training of both imputation and diffusion components. To assess its advantages, we compare three settings: (i) joint training with a brief TST pre-training (our approach), (ii) fully joint training from scratch, and (iii) a strict two-stage training where imputation is performed independently before generative training. Results in Tab. 15 demonstrate that our integrated approach consistently achieves the best discriminative performance across the Energy and Stock datasets, while fully joint training without pre-training suffers from unstable reconstructions, and the two-stage setup underperforms due to the lack of interaction between imputation and generative

Table 15: Average discriminative scores across 30%, 50%, and 70% drop-rates on the Energy and Stock datasets for sequence length of 768.

| Model | Energy | Stock |
|---|---|---|
| Joint training without pre-training | 0.278 | 0.076 |
| Training independently | 0.404 | 0.122 |
| **Joint training with pre-training (Ours)** | **0.222** | **0.024** |

learning. These findings highlight that even a short pre-training phase can stabilize learning and enable the model to effectively leverage imputed values during generation.

### E.5 Quantitative Evaluation - Full Results

We present the complete results, including standard deviations, for the experiment conducted in the main text in Sec. 5.2. In Tab. 18, Tab. 19, and Tab. 20, we provide detailed performance results across all missing rates of 30%, 50%, and 70% for sequence lengths of 24, 96, and 768, respectively.

### E.6 Qualitative Evaluation - Cont.

We provide the remaining missing rate analyses for the experiment described in Sec. 5.3. Fig. 8 presents the analysis for a 50% missing rate, while Fig. 9 shows the analysis for a 30% missing rate.

Additionally, we quantitatively assess the overlap between the original and generated data cloud points in the two-dimensional plane. We compute the Wasserstein distances between the original data and the generated samples. The results, presented in Table 16, indicate that our method consistently achieves the lowest Wasserstein distances across all missing rates and sequence lengths, outperforming KoVAE in every case. Notably, for long time series, our model exhibits significantly better performance, demonstrating its robustness in handling larger and more complex sequences with high missing rates. This underscores our approach's ability to closely match the true data distribution, even under challenging conditions, and its superior effectiveness in managing incomplete time series.

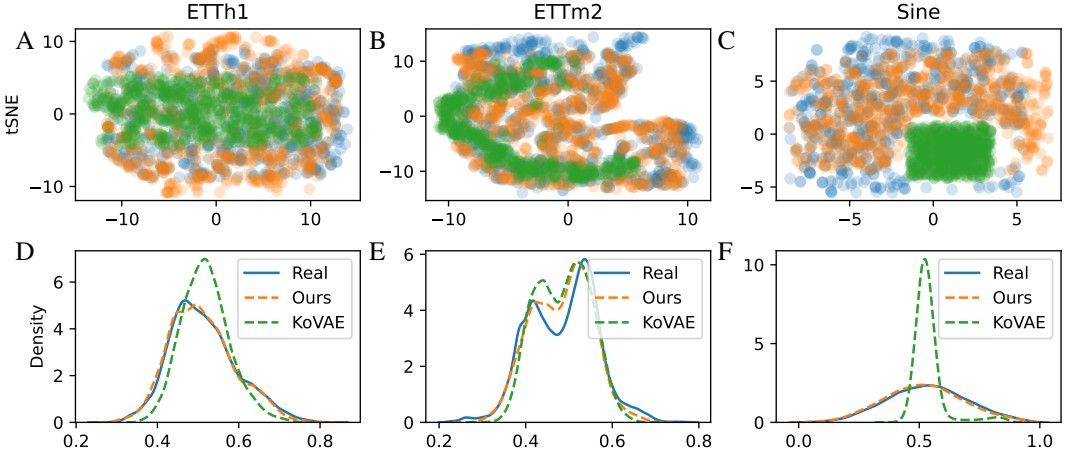

Figure 8: 2D t-SNE embeddings (top) and probability density functions (bottom) for the 50% missing rate on ETTh1 (short), ETTm2 (medium), and Sine (long) datasets.

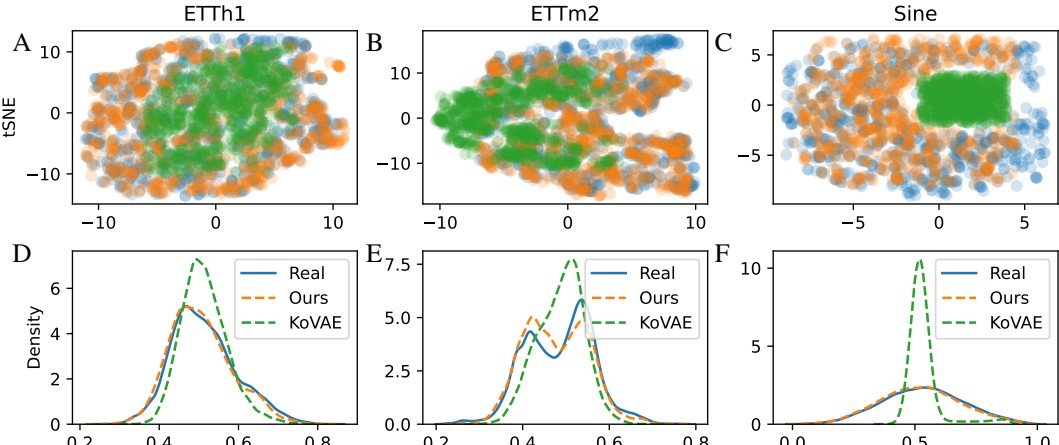

Figure 9: 2D t-SNE embeddings (top) and probability density functions (bottom) for the 30% missing rate on ETTh1 (short), ETTm2 (medium), and Sine (long) datasets.

Table 16: Wasserstein distances between original data clusters, generated samples, and clusters from KoVAE for various missing rates ($30\%, 50\%, 70\%$) and sequence lengths. Lower values indicate better similarity.

| Metric | Model | 30% Drop | | | 50% Drop | | | 70% Drop | | |
|---|---|---|---|---|---|---|---|---|---|---|
| | | Energy | Weather | Stocks | ETTh1 | ETTm2 | Sine | ETTh1 | ETTm2 | Sine |
| **Wass.↓** | KoVAE | 3.97 | 7.64 | 5.53 | 4.78 | 5.92 | 6.25 | 3.38 | 4.37 | 12.07 |
| | Ours | **1.08** | **2.85** | **1.68** | **1.62** | **1.84** | **1.50** | **1.33** | **2.39** | **2.42** |

### E.7 Complexity Analysis Cont.

Continuing from Sec. 5.1, where we summarized the upcoming experiment, we now present the full results.

In Table 17, we report the *net* training time (in hours) for our method (**Ours**) and **KoVAE** until convergence, measured under identical hardware (RTX3090) and batch size settings. Convergence was defined by the best discriminative score achieved during training; specifically, we sampled the generated data every 10 training epochs and computed the discriminative score against the real data. Note that the times shown exclude any overhead for data generation or evaluation; we only measure the pure training runtime until the point of highest discriminative performance.

These training times are presented for three different sequence lengths (24, 96, 768) and three missing rates (30%, 50%, 70%). To assess the relative speedup of our approach over KoVAE, we averaged the training times of each model (when valid entries were available), computed the percentage speedup as

$$\left( \frac{\text{KoVAE time} - \text{Ours time}}{\text{KoVAE time}} \times 100\% \right),$$

and then averaged these speedups across the three missing rates. Our results show that:

- At a sequence length of 24, **Ours** converges approximately $\sim 80\%$ faster on average.
- At a sequence length of 96, **Ours** converges approximately $\sim 87\%$ faster on average.
- At a sequence length of 768, **Ours** converges approximately $\sim 85\%$ faster on average.

These figures underscore the substantial reduction in training time provided by our method, ranging from about 80% to 90% faster than KoVAE in most settings, while still achieving superior performance based on the discriminative score.

Table 17: **Training Time** (in hours) for irregular time series across sequence lengths (24, 96, 768) and missing rates (30%, 50%, 70%).

| Seq. Len. | Drop % | Model | ETTh1 | ETTh2 | ETTm1 | ETTm2 | Weather | Energy | Sine | Stock |
|---|---|---|---|---|---|---|---|---|---|---|
| 24 | 30% | KoVAE | 5.00 | 15.08 | 11.65 | 7.49 | 9.23 | 12.20 | 5.78 | 2.71 |
| | | Ours | **0.73** | **0.61** | **1.18** | **0.90** | **4.77** | **2.80** | **0.50** | **0.37** |
| | 50% | KoVAE | 4.98 | **4.73** | 7.80 | 6.68 | 2.20 | 28.59 | 6.86 | 1.03 |
| | | Ours | **1.85** | 6.75 | **1.18** | **1.81** | **0.38** | **2.51** | **0.60** | **0.17** |
| | 70% | KoVAE | 9.45 | 12.55 | 10.98 | 5.48 | 6.08 | 8.91 | 4.22 | 2.36 |
| | | Ours | **1.27** | **0.92** | **0.52** | **1.58** | 5.82 | **2.02** | **0.35** | **0.08** |
| 96 | 30% | KoVAE | 25.80 | 8.10 | 13.79 | 25.92 | 33.87 | 4.94 | 6.90 | 7.71 |
| | | Ours | **3.38** | **0.80** | **1.35** | **0.95** | **2.43** | **2.36** | **1.20** | **0.23** |
| | 50% | KoVAE | 1.42 | 17.26 | 11.04 | 11.90 | 14.79 | 18.01 | 22.47 | 3.90 |
| | | Ours | **0.59** | **0.57** | **1.32** | **0.95** | **1.06** | **1.33** | **0.47** | **0.46** |
| | 70% | KoVAE | 31.91 | 7.12 | 19.90 | 15.77 | 17.52 | 18.11 | 2.91 | 7.76 |
| | | Ours | **0.61** | **4.34** | **1.05** | **3.65** | **2.12** | **1.49** | **2.72** | **1.60** |
| 768 | 30% | KoVAE | 62.19 | 59.70 | 92.48 | 69.48 | 40.88 | 13.30 | 18.03 | 16.59 |
| | | Ours | **4.72** | **1.96** | **7.57** | **6.45** | **16.46** | **3.50** | **2.93** | **3.11** |
| | 50% | KoVAE | 15.52 | 37.41 | 26.30 | 45.66 | 46.99 | 19.86 | 5.49 | 14.84 |
| | | Ours | **4.74** | **2.96** | **7.23** | **6.48** | **7.18** | **2.35** | **2.89** | **1.21** |

Table 18: Evaluation metrics for irregular time series with 24 sequence length (30%, 50%, 70% drop). Arrows (↑ / ↓) indicate whether higher or lower values are better.

| Metric | Model | Etth1 | Etth2 | Ettm1 | Ettm2 | Weather | Electricity | Energy | Sine | Stock | Mujoco |
|---|---|---|---|---|---|---|---|---|---|---|---|
| | | | | | | **30% Drop** | | | | | |
| Disc. ↓ | TimeGAN | 0.499 | 0.499 | 0.499 | 0.499 | 0.493 | 0.498 | 0.448 | 0.494 | 0.463 | 0.471 |
| | GT-GAN | 0.473 | 0.371 | 0.420 | 0.369 | 0.472 | 0.409 | 0.333 | 0.363 | 0.251 | 0.249 |
| | KoVAE | 0.208 | 0.075 | 0.045 | 0.077 | 0.229 | 0.497 | 0.280 | 0.035 | 0.162 | 0.123 |
| | Ours | **0.020** | **0.009** | **0.014** | **0.006** | **0.029** | **0.399** | **0.048** | **0.013** | **0.007** | **0.009** |
| Pred. ↓ | TimeGAN | 0.156 | 0.305 | 0.146 | 0.262 | 0.388 | 0.183 | 0.375 | 0.145 | 0.087 | 0.118 |
| | GT-GAN | 0.174 | 0.092 | 0.119 | 0.097 | 0.147 | 0.148 | 0.066 | 0.099 | 0.021 | 0.048 |
| | KoVAE | 0.058 | 0.050 | **0.044** | 0.051 | 0.029 | **0.048** | 0.049 | 0.074 | 0.019 | 0.043 |
| | Ours | **0.052** | **0.043** | **0.044** | **0.045** | **0.022** | **0.048** | **0.047** | **0.070** | **0.012** | **0.040** |
| Fid. ↓ | TimeGAN | 2.934 | 2.565 | 2.437 | 2.924 | 1.612 | 18.04 | 4.440 | 2.919 | 2.475 | 3.628 |
| | GT-GAN | 1.689 | 15.26 | 27.43 | 6.902 | 1.161 | 9.907 | 1.305 | 1.810 | 2.429 | 0.656 |
| | KoVAE | 1.769 | 0.211 | 0.181 | 0.609 | 0.539 | 7.606 | 0.645 | 0.048 | 0.741 | 0.428 |
| | Ours | **0.071** | **0.023** | **0.023** | **0.010** | **0.018** | **3.451** | **0.033** | **0.032** | **0.009** | **0.028** |
| Corr. ↓ | TimeGAN | 6.317 | 0.862 | 2.290 | 0.357 | 0.744 | 11.13 | 3.663 | 2.131 | 0.273 | 0.844 |
| | GT-GAN | 7.167 | 0.918 | 2.519 | 0.358 | 0.782 | 14.93 | 3.855 | 3.141 | 0.264 | 0.803 |
| | KoVAE | 0.148 | 0.088 | 0.162 | 0.483 | 1.852 | 4.351 | 2.910 | 0.049 | 0.032 | 0.561 |
| | Ours | **0.070** | **0.048** | **0.059** | **0.045** | **0.424** | **2.041** | **0.815** | **0.016** | **0.007** | **0.331** |
| | | | | | | **50% Drop** | | | | | |
| Disc. ↓ | TimeGAN | 0.499 | 0.499 | 0.499 | 0.499 | 0.499 | 0.498 | 0.479 | 0.496 | 0.487 | 0.483 |
| | GT-GAN | 0.462 | 0.371 | 0.407 | 0.376 | 0.496 | 0.391 | 0.317 | 0.372 | 0.265 | 0.270 |
| | KoVAE | 0.188 | 0.086 | 0.057 | 0.077 | 0.498 | 0.499 | 0.298 | 0.030 | 0.092 | 0.117 |
| | Ours | **0.032** | **0.005** | **0.013** | **0.013** | **0.035** | **0.360** | **0.065** | **0.014** | **0.007** | **0.007** |
| Pred. ↓ | TimeGAN | 0.210 | 0.343 | 0.157 | 0.292 | 0.404 | 0.230 | 0.501 | 0.123 | 0.058 | 0.402 |
| | GT-GAN | 0.176 | 0.091 | 0.118 | 0.096 | 0.127 | 0.064 | 0.101 | 0.018 | 0.056 |
| | KoVAE | 0.057 | 0.053 | 0.045 | 0.050 | 0.114 | **0.043** | 0.050 | 0.072 | 0.019 | 0.042 |
| | Ours | **0.053** | **0.045** | **0.043** | **0.041** | **0.022** | 0.049 | **0.047** | **0.068** | **0.012** | **0.040** |
| Fid. ↓ | TimeGAN | 4.131 | 3.132 | 2.642 | 2.693 | 2.839 | 20.184 | 6.408 | 2.124 | 2.352 | 4.141 |
| | GT-GAN | 1.504 | 5.839 | 4.201 | 9.468 | 5.919 | 9.741 | 1.935 | 1.785 | 2.258 | 0.664 |
| | KoVAE | 1.309 | 0.319 | 0.207 | 0.115 | 9.830 | 3.972 | 0.421 | 0.030 | 0.225 | 0.371 |
| | Ours | **0.134** | **0.026** | **0.045** | **0.018** | **0.040** | **3.633** | **0.061** | **0.007** | **0.057** | **0.026** |
| Corr. ↓ | TimeGAN | 2.293 | 0.932 | 1.864 | 0.352 | 0.835 | 13.79 | 3.761 | 2.192 | 2.021 | 0.825 |
| | GT-GAN | 7.088 | 0.928 | 2.443 | 0.361 | 0.807 | 14.91 | 3.971 | 3.204 | 0.255 | 0.804 |
| | KoVAE | 0.173 | 0.283 | 0.132 | 0.157 | 5.027 | 4.152 | 1.822 | 0.029 | 0.072 | 0.555 |
| | Ours | **0.112** | **0.047** | **0.058** | **0.026** | **0.363** | **2.044** | **0.872** | **0.015** | **0.011** | **0.342** |
| | | | | | | **70% Drop** | | | | | |
| Disc. ↓ | TimeGAN | 0.500 | 0.499 | 0.500 | 0.500 | 0.499 | 0.500 | 0.496 | 0.500 | 0.488 | 0.494 |
| | GT-GAN | 0.478 | 0.366 | 0.409 | 0.353 | 0.475 | 0.480 | 0.325 | 0.278 | 0.230 | 0.275 |
| | KoVAE | 0.196 | 0.081 | 0.048 | 0.046 | 0.269 | 0.498 | 0.392 | 0.065 | 0.101 | 0.119 |
| | Ours | **0.058** | **0.013** | **0.010** | **0.015** | **0.106** | **0.392** | **0.128** | **0.008** | **0.010** | **0.009** |
| Pred. ↓ | TimeGAN | 0.436 | 0.359 | 0.401 | 0.387 | 0.390 | 0.372 | 0.496 | 0.734 | 0.072 | 0.442 |
| | GT-GAN | 0.207 | 0.094 | 0.137 | 0.089 | 0.162 | 0.149 | 0.076 | 0.088 | 0.020 | 0.051 |
| | KoVAE | 0.056 | 0.060 | 0.046 | 0.048 | 0.028 | 0.051 | 0.052 | 0.076 | **0.012** | 0.044 |
| | Ours | **0.054** | **0.049** | **0.045** | **0.047** | **0.023** | **0.049** | **0.047** | **0.069** | **0.012** | **0.041** |
| Fid. ↓ | TimeGAN | 2.356 | 3.900 | 5.179 | 4.036 | 2.683 | 31.95 | 8.674 | 3.296 | 3.178 | 4.432 |
| | GT-GAN | 3.442 | 4.805 | 11.24 | 2.784 | 1.194 | 10.33 | 1.354 | 1.498 | 1.857 | 0.671 |
| | KoVAE | 1.477 | 0.215 | 0.153 | 0.116 | 0.727 | 7.610 | 0.820 | 0.033 | 0.141 | 0.292 |
| | Ours | **0.167** | **0.056** | **0.073** | **0.043** | **0.451** | **3.653** | **0.301** | **0.007** | **0.041** | **0.044** |
| Corr. ↓ | TimeGAN | 6.821 | 0.959 | 2.470 | 0.387 | 0.728 | 14.80 | 3.848 | 3.051 | 3.354 | 0.867 |
| | GT-GAN | 7.190 | 0.921 | 2.438 | 0.351 | 0.785 | 14.92 | 3.842 | 3.502 | 0.254 | 0.809 |
| | KoVAE | 0.228 | 0.160 | 0.096 | 0.144 | 1.817 | 4.213 | 3.157 | 0.029 | 0.095 | 0.566 |
| | Ours | **0.071** | **0.104** | **0.079** | **0.055** | **0.403** | **2.007** | **1.330** | **0.015** | **0.037** | **0.348** |

Table 19: Evaluation metrics for irregular time series with 96 sequence length (30%, 50%, 70% drop). Arrows (↑ / ↓) indicate whether higher or lower values are better.

| Metric | Model | Etth1 | Etth2 | Ettm1 | Ettm2 | Weather | Energy | Sine | Stock |
|---|---|---|---|---|---|---|---|---|---|
| | | | | | **30% Drop** | | | | |
| Disc. ↓ | KoVAE | 0.255 | 0.096 | 0.264 | 0.075 | 0.290 | 0.416 | 0.244 | 0.114 |
| | Ours | **0.037** | **0.036** | **0.033** | **0.028** | **0.084** | **0.130** | **0.004** | **0.011** |
| Pred. ↓ | KoVAE | 0.062 | 0.062 | 0.053 | 0.047 | 0.040 | 0.077 | 0.164 | 0.016 |
| | Ours | **0.052** | **0.045** | **0.044** | **0.042** | **0.023** | **0.048** | **0.155** | **0.010** |
| Fid ↓ | KoVAE | 5.223 | 0.915 | 4.073 | 0.645 | 2.942 | 4.075 | 2.725 | 0.944 |
| | Ours | **0.156** | **0.112** | **0.147** | **0.060** | **0.169** | **0.193** | **0.018** | **0.110** |
| Corr. ↓ | KoVAE | 0.201 | 0.301 | 0.177 | 0.203 | 2.730 | 4.677 | 0.058 | 0.086 |
| | Ours | **0.102** | **0.075** | **0.087** | **0.042** | **0.306** | **0.827** | **0.017** | **0.016** |
| | | | | | **50% Drop** | | | | |
| Disc. ↓ | KoVAE | 0.304 | 0.077 | 0.290 | 0.114 | 0.358 | 0.321 | 0.188 | 0.111 |
| | Ours | **0.070** | **0.067** | **0.047** | **0.017** | **0.190** | **0.153** | **0.003** | **0.018** |
| Pred. ↓ | KoVAE | 0.063 | 0.055 | 0.053 | 0.054 | 0.051 | 0.063 | 0.161 | 0.016 |
| | Ours | **0.054** | **0.053** | **0.044** | **0.046** | **0.025** | **0.048** | **0.155** | **0.011** |
| Fid ↓ | KoVAE | 5.370 | 0.781 | 4.663 | 1.325 | 4.117 | 4.955 | 2.334 | 1.003 |
| | Ours | **0.516** | **0.487** | **0.179** | **0.130** | **0.567** | **0.182** | **0.018** | **0.105** |
| Corr. ↓ | KoVAE | 0.246 | 0.288 | 0.247 | 0.638 | 2.490 | 5.142 | 0.044 | 0.085 |
| | Ours | **0.154** | **0.219** | **0.090** | **0.113** | **1.176** | **1.186** | **0.019** | **0.011** |
| | | | | | **70% Drop** | | | | |
| Disc. ↓ | KoVAE | 0.294 | 0.137 | 0.274 | 0.079 | 0.374 | 0.332 | 0.286 | 0.073 |
| | Ours | **0.102** | **0.057** | **0.039** | **0.044** | **0.182** | **0.272** | **0.002** | **0.021** |
| Pred. ↓ | KoVAE | 0.062 | 0.055 | 0.056 | 0.049 | 0.047 | 0.064 | 0.171 | 0.029 |
| | Ours | **0.053** | **0.050** | **0.046** | **0.044** | **0.027** | **0.051** | **0.155** | **0.012** |
| Fid ↓ | KoVAE | 6.932 | 1.638 | 3.473 | 1.019 | 3.983 | 5.051 | 8.083 | 0.657 |
| | Ours | **0.399** | **0.926** | **0.187** | **0.235** | **0.447** | **0.553** | **0.016** | **0.112** |
| Corr. ↓ | KoVAE | 0.226 | 0.496 | 0.101 | 0.465 | 2.607 | 4.611 | 0.565 | 0.095 |
| | Ours | **0.087** | **0.159** | **0.100** | **0.126** | **1.187** | **1.470** | **0.013** | **0.015** |

Table 20: Evaluation metrics for irregular time series with 768 sequence length (30%, 50%, 70% drop). Arrows (↑ / ↓) indicate whether higher or lower values are better.

| Metric | Model | Etth1 | Etth2 | Ettm1 | Ettm2 | Weather | Energy | Sine | Stock |
|--------|-------|-------|-------|-------|-------|---------|--------|------|-------|
| | | | | | **30% Drop** | | | | |
| Disc. ↓ | KoVAE | 0.239 | 0.237 | 0.282 | 0.160 | 0.411 | 0.371 | 0.284 | 0.289 |
| | Ours | **0.045** | **0.032** | **0.067** | **0.047** | **0.093** | **0.145** | **0.005** | **0.025** |
| Pred. ↓ | KoVAE | 0.077 | 0.074 | 0.059 | 0.081 | 0.061 | 0.089 | 0.223 | 0.020 |
| | Ours | **0.052** | **0.056** | **0.047** | **0.051** | **0.027** | **0.025** | **0.204** | **0.011** |
| Fid ↓ | KoVAE | 11.16 | 9.448 | 11.47 | 8.869 | 12.21 | 25.50 | 38.71 | 7.431 |
| | Ours | **0.318** | **0.258** | **0.373** | **0.220** | **0.110** | **0.755** | **0.184** | **0.116** |
| Corr. ↓ | KoVAE | 0.378 | 0.528 | 0.480 | 0.859 | 3.136 | 8.138 | 0.411 | 0.041 |
| | Ours | **0.088** | **0.119** | **0.126** | **0.086** | **0.269** | **0.849** | **0.006** | **0.004** |
| | | | | | **50% Drop** | | | | |
| Disc. ↓ | KoVAE | 0.270 | 0.191 | 0.197 | 0.225 | 0.428 | 0.372 | 0.426 | 0.302 |
| | Ours | **0.061** | **0.030** | **0.061** | **0.048** | **0.097** | **0.244** | **0.009** | **0.029** |
| Pred. ↓ | KoVAE | 0.074 | 0.064 | 0.056 | 0.084 | 0.064 | 0.086 | 0.222 | 0.023 |
| | Ours | **0.053** | **0.057** | **0.047** | **0.048** | **0.027** | **0.051** | **0.204** | **0.015** |
| Fid ↓ | KoVAE | 14.56 | 7.412 | 11.51 | 9.373 | 18.59 | 19.58 | 38.49 | 8.274 |
| | Ours | **0.589** | **0.248** | **0.305** | **0.211** | **0.225** | **0.916** | **0.211** | **0.199** |
| Corr. ↓ | KoVAE | 0.290 | 0.574 | 0.428 | 0.842 | 4.836 | 12.63 | 0.336 | 0.085 |
| | Ours | **0.103** | **0.128** | **0.107** | **0.119** | **0.455** | **1.473** | **0.006** | **0.042** |
| | | | | | **70% Drop** | | | | |
| Disc. ↓ | KoVAE | 0.206 | 0.176 | 0.231 | 0.203 | 0.445 | 0.409 | 0.340 | 0.263 |
| | Ours | **0.160** | **0.072** | **0.046** | **0.061** | **0.116** | **0.251** | **0.005** | **0.013** |
| Pred. ↓ | KoVAE | 0.065 | 0.071 | 0.065 | 0.062 | 0.086 | 0.088 | 0.234 | 0.051 |
| | Ours | **0.054** | **0.056** | **0.047** | **0.052** | **0.028** | **0.050** | **0.204** | **0.013** |
| Fid ↓ | KoVAE | 16.05 | 8.052 | 17.54 | 6.595 | 21.69 | 28.46 | 38.60 | 6.114 |
| | Ours | **1.739** | **0.812** | **0.496** | **0.362** | **0.619** | **0.718** | **0.317** | **0.167** |
| Corr. ↓ | KoVAE | 0.331 | 0.718 | 0.305 | 0.515 | 1.786 | 5.49 | 0.391 | **0.013** |
| | Ours | **0.118** | **0.071** | **0.133** | **0.179** | **0.650** | **0.798** | **0.006** | 0.034 |

