# OpenReview forum: "A Diffusion Model for Regular Time Series Generation from Irregular Data with Completion and Masking"
_NeurIPS.cc/2025/Conference — NeurIPS 2025 poster_

### Official Review · Reviewer_ySTA · 2025-06-23

**Clarity:** 3
**Significance:** 3
**Originality:** 3
**Rating:** 5
**Confidence:** 4

**Summary:**

This paper propose a novel two-step framework for generating realistic time series data. First, a time series transformer is employed to complete irregular sequences. Second, a vision-based diffusion model with masking minimizes dependence on the completed values. The proposed method achieves state-of-the-art performance, achieving significant improvement in discriminative score and computational cost . Overall, I feel this paper is well-written and of good quality. I have some concerns below, which I believe can be addressed. I am happy to raise my score if the authors approach them properly.

**Questions:**

Q1: The discriminative score typically depends on the choice of classifier used to distinguish real from synthetic data. As shown in [1], only a strong classifier can effectively measure fidelity. Could the authors provide more details about the classifier used to compute this score?

Q2: In Table 5, when the noise level increases from 0.1 to 0.15, the predictive performance exhibits significant degradation. This is somehow weird that the increase of noise is not that large but causes significant deterioration. It would be better to provide some explanination for this.

Q3: Existing time series generation methods also include the combination of VAE and Diffusion. How is the method compared with these?

Q4: Can the method be applied to time-series data with categorical and numerical attributes?









**References:**

[1] Tao, L., Xu, S., Wang, C.H., Suh, N. and Cheng, G., 2024. Discriminative estimation of total variation distance: A fidelity auditor for generative data. arXiv preprint arXiv:2405.15337.

**Ethical Concerns:**

["NO or VERY MINOR ethics concerns only"]

**Final Justification:**

After the rebuttal, the authors had greatly addressed my concerns and I think this is a good paper.

**Limitations:**

yes

**Quality:**

3

**Strengths And Weaknesses:**

**Strengths:**

1. Overall, this paper is well-written and easy to follow.

2. The paper conducted extensive experiments across 11 datasets and 12 tasks demonstrate state-of-the-art performance, with up to 70% improvement in discriminative score.

3. The paper includes insightful ablation studies and toy examples to demonstrate the importance of natural neighborhoods.

4. The main contribution of the proposed method is combining completion via a time series transformer with masking during training of a vision-based diffusion model, mitigating the issue of unnatural neighborhoods caused by zero-filled missing values.


**Weaknesses:**

1. The comparison is not thorough. The authors may include some VAE-Diffusion methods for comparison.

2. The paper leans heavily on engineering contributions. It would benefit from a stronger theoretical foundation or more detailed justification for design choices. For example, why the proposed combination of TST and masking is particularly well-suited to the task compared to other alternatives.

---

> ### Author Rebuttal · Authors · 2025-07-30
>
> 1. **“The comparison is not thorough. The authors may include some VAE-Diffusion methods for comparison.”**
>
>     To the best of our knowledge, only two existing methods combine VAE and diffusion for unconditional time-series generation: TimeLDM and TimeAutoDiff. Since TimeLDM does not provide public code, we were unable to include it in our experiments.
>
>     To directly adress you concern, we have conducted experemnation with TimeAutoDiff + TST as a representative baseline. In this setup, we replaced the ImagenTime generator in our pipeline with TimeAutoDiff, and used TST to impute missing values prior to generation. The model was trained and evaluated under the exact same protocol as ours. To ensure the performance gains of our method are not tied to a specific generative architecture, we further included TransFusion + TST, a strong diffusion model that operates directly on time-series data without relying on image representations. This helps isolate the impact of our contributions from the backbone architecture itself. We attach below the table results. We will add this new experimentation to the final revision.
>
>     As shown in Tab. 1 below, both TimeAutoDiff + TST and TransFusion + TST underperform significantly across all datasets, sequence lengths, and missing rates. These results highlight that our improvements do not stem from using a more powerful generator or imputer, but from our core innovation: combining natural completions with a masked loss applied only to observed values. This principled design leads to consistent and substantial gains over all tested alternatives.
>
>
>     **Table 1: Avg. Disc. Scores @ 30/50/70% drop (Len=24/96/768). E = Energy, S=Stock**
>
>     | Model              | E30  | S30  | E50  | S50  | E70  | S70  |
>     |-------------------|------|------|------|------|------|------|
>      | KoVAE + TST       | 0.340 | 0.173 | 0.360 | 0.176 | 0.403 | 0.139 |
>     | TimeAutoDiff + TST| 0.297 | 0.103 | 0.333 | 0.102 | 0.465 | 0.420 |
>     | TransFusion + TST | 0.291 | 0.082 | 0.336 | 0.092 | 0.439 | 0.104 |
>     | Ours (Mask Only)  | 0.361 | 0.170 | 0.347 | 0.294 | 0.429 | 0.372 |
>     | Ours (No Mask)    | 0.308 | 0.028 | 0.379 | 0.073 | 0.420 | 0.066 |
>     | **Ours**          | **0.116** | **0.014** | **0.154** | **0.019** | **0.217** | **0.014** |
>
>
> ---
>
> 2. **“..It would benefit from a stronger theoretical foundation or more detailed justification for design choices. For example, why the proposed combination of TST and masking is particularly well-suited to the task..”**
>
>     We appreciate the reviewer’s suggestion to further justify our design choices. Our architectural decisions are grounded in detailed empirical analysis and targeted error investigations of prior approaches, which revealed two key limitations:
>
>     - NCDE-Based Imputation Challenges: Prior work that utilizes NCDE for imputation, such as KoVAE, suffers from scalability issues—NCDE becomes computationally infeasible for long sequences and introduces substantial overhead even for short to mid-length time series.
>
>     - Lack of Masking in Imputation Pipelines: Methods that omit masking during the imputation process (e.g., using a deterministic completion step as input to diffusion) risk encoding inaccurate or biased reconstructions. The diffusion model in such cases learns from data distributions that do not reflect the underlying true data, which leads to compounding errors during generation.
>
>     Our ablation studies at Tab 1. below directly support these observations. Specifically:
>
>     - Masking-Only and Ours (W/O Masking) configurations fail to accurately capture the ground-truth distribution, underscoring the importance of masking during diffusion.
>
>     - Simply replacing NCDE with TST in KoVAE improves computational efficiency but does not significantly improve performance, highlighting that the choice of completion model alone is not sufficient.
>
>     - Even when employing strong diffusion models like TransFusion with TST-based completion, performance degrades notably without masking.
>
>     In contrast, our proposed architecture-combining TST completion with strategic masking during diffusion-achieves clear gains across all tasks. This combination allows the diffusion model to model uncertainty effectively and prevents over-reliance on potentially inaccurate imputations. Our findings thus demonstrate that the interplay between completion and masking is critical, and neither component alone explains the performance improvements. This design is not arbitrary but directly motivated by both theoretical considerations (avoiding error accumulation via inaccurate inputs) and empirical results.
>
> ---
>
> 3. **“...Could the authors provide more details about the classifier used to compute this score?”**
>
>     To ensure **fair and meaningful comparisons with prior work**, we used the same pre-trained classifier employed by baseline methods such as GT-GAN and KoVAE.
>
>     This classifier is a task-specific model trained to distinguish between real and synthetic sequences using supervised learning on labeled data. In our case, the classifier was trained exclusively on real and ground-truth-labeled data and then evaluated on a balanced test set containing both real and generated samples.
>
>     To evaluate its fidelity-measuring capability, we ensured that the classifier:
>
>     - Shows near-zero discriminative power when evaluated on real vs. real samples drawn from the same distribution (e.g., original data vs. shuffled batches), confirming that the classifier does not detect spurious differences in such control setups.
>     - Demonstrates consistent performance across multiple random seeds and datasets.
>     - Is not overfitting to a specific generation method, but generalizes well across different types of synthetic data.
>
> ---
>
> 4. **“In Table 5, when the noise level increases from 0.1 to 0.15, the predictive performance exhibits significant degradation. This is somehow weird that the increase of noise is not that large but causes significant deterioration...”**
>
>     Thank you for pointing this out. We agree that at first glance, increasing the noise level from 0.1 to 0.15 may seem like a relatively small change. However, the impact on predictive performance is more pronounced due to the scale of the underlying data.
>
>     In our experiments, the noise was added as additive Gaussian noise sampled from a normal distribution N(0, σ), where σ corresponds to the specified noise level (e.g., 0.1 or 0.15). Importantly, this noise is added independently of the original data distribution or scale.
>
>
>     For example, in the Energy dataset the standard deviation of the target variable is approximately 0.22. Therefore, adding noise sampled from N(0, 0.15) introduces noise with a standard deviation that is about 68% of the data's inherent variability. This means that the relative magnitude of the noise is substantial and not minor, as the absolute values might suggest.
>
>     Thus, the observed degradation in predictive performance is consistent with the relatively large proportion of noise added compared to the signal. We will clarify this point in the revised manuscript for better transparency.
>
> ---
>
> 5. **“Existing time series generation methods also include the combination of VAE and Diffusion. How is the method compared with these?”**
>
>     Answered in 1.
>
> 6. **“Can the method be applied to time-series data with categorical and numerical attributes?”**
>
>     In our paper, we followed benchmarks from previous work, which focused on numerical datasets only. To improve evaluation and show broader applicability, we added 7 new datasets and 4 sequence lengths. In response to your question, we further extended our method to support mixed-type time series with both categorical and numerical attributes. Our approach treats categorical variables using learnable embeddings that are jointly optimized during training. Specifically, we replace each categorical feature with a dense embedding vector, transforming the input from mixed data types into a unified continuous representation. For example, a time series with 5 numerical features and 3 categorical features becomes a fully continuous representation where categorical indices are mapped to learned embedding vectors that are concatenated with the numerical features.
>     The key advantage is that these embeddings are learned end-to-end with the generative model, allowing the model to find the optimal representations for categorical values that best capture their relationships within the temporal context. The diffusion model then operates entirely in this continuous embedding space, naturally handling both numerical dynamics and categorical dependencies.
>     During generation, we apply a simple postprocessing step to map the generated embedding vectors back to discrete categories by finding the nearest learned embedding or using classification heads, depending on the setup.
>
>     As part of this extension, we evaluated our method on the Traffic dataset—which includes both categorical and numerical features—to demonstrate this capability [1]. As shown in Tab. 2 below, our method achieved the best discriminative scores across all missing rates.
>
>     [1] Hogue, J. (2019). Metro Interstate Traffic Volume [Dataset]. UCI Machine Learning Repository. https://doi.org/10.24432/C5X60B.
>
>     **Table 2: Disc. scores on Traffic @ 30/50/70% drop (Len=24, 96).**
>
>     | Seq Len | Model  | 30%   | 50%   | 70%   |
>     |---------|--------|--------|--------|--------|
>     | 24      | GT-GAN | 0.481  | 0.473  | 0.485  |
>     |         | KoVAE  | 0.154  | 0.172  | 0.222  |
>     |         | **Ours** | **0.061** | **0.064** | **0.087** |
>     | 96      | GT-GAN | 0.493  | 0.491  | 0.488  |
>     |         | KoVAE  | 0.212  | 0.245  | 0.307  |
>     |         | **Ours** | **0.073** | **0.091** | **0.102** |

---

> > ### Author Response · Authors · 2025-08-03
> >
> > We would like to gently remind you that we have submitted our rebuttal, which directly addresses the key concerns you raised.
> > If anything remains unclear, please let us know — your feedback is highly valued.
> > Thank you again for reviewing our work.

---

> > > ### Comment · Reviewer_ySTA · 2025-08-04
> > > **Response to the authors**
> > >
> > > I would like to thanks the reviewers for the detailed response. I think you significantly address my concerns. I will consider raising my rating and support your paper.

---

### Official Review · Reviewer_KWFu · 2025-06-24

**Clarity:** 3
**Significance:** 3
**Originality:** 2
**Rating:** 4
**Confidence:** 3

**Summary:**

This paper proposes a generative modeling method based on ImagenTime that can handle irregular time series data.

**Questions:**

1. There are two types of irregular data that I understand. The first is the data with missing values mentioned by the author, and the other is the data with irregular sampling frequency. Does the author's method seem to only address the first scenario?

2. When applying the design of this paper to sequences without missing values, may it lead to performance degradation compared to the original ImagenTime?

3. If I understand correctly, it seems that this paper mainly combines KoVAE's stage1 with ImagenTime, am I right?

4.  I am confused about the experimental setup of this type of method. Generally speaking, the absence of time series in reality has a physical mechanism, such as the possibility of wind power sequences experiencing extreme weather leading to downtime, and so on. Does using random masking directly to simulate missing values really have practical value in the real physical world? In addition, the simulation of missing values has a pre-defined length? Becasuse many missing values occur continuously. If they are only missing at certain time steps, it will be easier to solve them through imputation.

5. Can the author provide some relevant experiments to prove the advantages of the integrated setting to the two-stage setting that first performing imputation before generating tasks?

**Ethical Concerns:**

["NO or VERY MINOR ethics concerns only"]

**Final Justification:**

The authors clarify most of my concerns including the novelty and difference between the work and ImagenTime.

**Limitations:**

Yes

**Quality:**

3

**Strengths And Weaknesses:**

Strengths

1. Superior performance: performs well on multiple generation tasks and datasets, significantly outperforming existing methods

2. High efficiency: Compared to traditional methods, it significantly reduces computational costs and has good practicality

Weaknesses

1. The main improvement is based on ImagenTime to adapt to irregular time series (specifically referring to missing values in normal time series). It feels that contribution is actually very marginal because most of it is from ImagenTime.

---

> ### Author Rebuttal · Authors · 2025-07-30
>
> 1. **“...It feels that contribution is actually very marginal because most of it is from ImagenTime.”**
>
>     ImagenTime was not designed to handle irregular time series. It assumes fully observed, regularly sampled data, and does not include any mechanism for dealing with missing values or imputation uncertainty. When naively applied to irregular data—either by zero-filling or by completing missing values and training on them as samples from the true distribution—performance degrades significantly.
>
>     To make this concrete, we include an ablation in **Tab. 1 in the vxoC reviewer's response** (Tab. 6 in the paper) comparing several variants. The setup labeled “Ours (W/O masking)” completes the missing values using TST and trains ImagenTime exactly as in the original paper, with no modification to the loss function. The “Ours (Mask Only)” setup fills missing values with zeros and applies masking in the loss. Both approaches perform significantly worse than our full method.
>
>     Specifically, on average across all sequence lengths and missing rates:
>     - Our approach improves the **discriminative score by 64.6% on the Stocks dataset** and **56.3% on the Energy dataset** compared to Ours (W/O Masking).
>     - Compared to Ours (Mask Only), the improvements are **94.0% on Stocks** and **58.3% on Energy**.
>
>     Each of these baselines reveals a different failure mode:
>
>     - In Ours (W/O Masking), the model is trained to match the completed values as if they were true samples. This leads to **error propagation**—any mistake made during imputation is treated as supervision, degrading generalization and generation quality.
>     - In Ours (Mask Only), missing values are replaced with zeros and masked out in the loss. While this prevents error propagation, the model is exposed to **highly unnatural local neighborhoods**, which weakens its ability to learn realistic generative patterns.
>
>     Our method avoids both pitfalls: the model sees **imputed values that form natural, coherent neighborhoods**, but the **loss is computed only on observed values**. This allows the model to benefit from contextual structure while remaining robust to imputation noise—a key factor in the performance gains we observe.
>
>     This ablation decisively shows that the performance gain does not stem from ImagenTime, or any other time-series generator, but from our **specific novel combination** of natural neighbors with masked loss on ground truth data only.
>
>     See also App. B.2 for additional details on our loss function.
>
> ---
>
> 2. **“There are two types of irregular data that I understand. The first is the data with missing values, and the other is the data with irregular sampling frequency. Does the author's method seem to only address the first scenario?”**
>
>     Our method can address both types of irregularity: missing values and irregular sampling frequency. While we focused our evaluation on missing values, your second scenario can also be accommodated, as we describe below.
>
>     In the case of irregular sampling frequency, we first define a regular time grid by selecting a fixed interval that matches the finest granularity of interest for the application (e.g., 1 second, 1 minute, etc.). This grid redefines the time axis as evenly spaced, and any time points without measurements are treated as missing values (NaNs).
>
>     For example, if the original data contains observations at times *t = 1, 4, 6*, and we choose a fixed interval of 1, we construct a grid:
>     *t = 1, 2, 3, 4, 5, 6*.
>     We then place the known values at their respective times, and treat *t = 2, 3, 5* as missing. This allows us to apply our method directly, without modification.
>
>     In summary, by transforming irregular timestamps into a regularly sampled frame with explicit missing entries, our framework can be extended to handle both missingness and irregular temporal spacing in a unified way.
>
> ---
>
> 3. **“When applying the design of this paper to sequences without missing values, may it lead to performance degradation compared to the original ImagenTime?”**
>
>     When the input data has no missing values, our system naturally **bypasses the completion step and masking**, since all values come from the true distribution. In this case, we simply train ImagenTime as-is, without any modification to its architecture or loss.
>
>     Therefore, **no performance degradation is expected** in the fully observed setting. Our framework is designed to be adaptive: it reduces to ImagenTime in the absence of missingness and enhances it when irregularities are present.
>
> ---
>
> 4. **“If I understand correctly, it seems that this paper mainly combines KoVAE's stage1 with ImagenTime, am I right?”**
>
>     While both our method and KoVAE use a two-stage structure, **our approach differs fundamentally**:
>
>     - KoVAE uses NCDE completion, which is slow, memory-intensive, and difficult to scale.
>     - We replace this with a **lightweight and scalable TST module** that runs efficiently on longer sequences and higher-dimensional datasets (**see our responses to Reviewers vxoC and zmvt for benchmarks**).
>
>     More importantly, the performance gain in our method is **not simply due to using TST**. The key innovation is the **combination of natural neighbors with a masked diffusion loss**:
>
>     - Unlike KoVAE, we **never treat completed values as ground truth**.
>     - The completed values serve as **context only**, and the **loss is computed solely on the observed entries**.
>
>     This distinction is empirically supported: when we evaluate KoVAE + TST (i.e., using TST for completion but without masking), the performance remains significantly lower than our full method (see Tab. 1 in zmvt for full results).
>
> ---
>
> 5. **“...the absence of time series in reality has a physical mechanism. Does using random masking directly to simulate missing values really have practical value in the real physical world? In addition, the simulation of missing values has a pre-defined length? Because many missing values occur continuously. If they are only missing at certain time steps, it will be easier to solve them through imputation.”**
>
>     To ensure fair comparison, we follow the **standard protocol** adopted in prior works such as KoVAE and GT-GAN, where missing values are randomly dropped from regular sequences. While it is true that in some cases missingness is driven by physical processes (e.g., sensor downtime), many real-world time series exhibit **irregular, sparse, or stochastic missingness patterns**.
>
>     For instance, in the ERA5 reanalysis dataset (Hersbach et al., 2020), which records temperature over large spatial grids (e.g., 80×80 locations across 120 time steps), missing values frequently occur not due to contiguous outages, but because of sensor availability, asynchronous updates, and communication errors. These lead to random and non-continuous missingness across both space and time.
>
>     That said, our method is **not limited** to random masking patterns:
>
>     - The TST- completion and masked loss mechanism apply equally well to block-wise or structured missingness.
>     - At high masking rates (e.g. 70%), long consecutive NaN segments naturally emerge, even under random masking-effectively simulating structured missingness patterns.
>
>     To address your concern more directly, we ran an additional experiment comparing our model’s performance under two missingness types:
>
>     - (i) 40% randomly dropped values, and
>     - (ii) 40% continuous missing blocks.
>
>     This experiment was conducted on the Weather & Energy datasets using sequence lengths of 24 and 96. The results (see Tab. 1 below) show comparable performance under both settings—with some cases slightly favoring random drop, and others favoring block-missingness. These results support the robustness of our method across a range of missingness patterns.
>
>     **Table 1: Disc./Pred. scores (↓) for 40% random vs. continuous missingness on Weather (W in the table) & Energy (E in the table) (Len=24, 96).**
>
>     | Len | Metric | W-Rand | W-Cont | E-Rand | E-Cont |
>     |-----|--------|--------|--------|--------|--------|
>     | 24  | Disc.  | 0.057  | **0.053** | **0.081** | 0.082  |
>     |     | Pred.  | **0.021** | 0.025  | 0.047  | **0.043** |
>     | 96  | Disc.  | **0.154** | 0.165  | **0.185** | 0.191  |
>     |     | Pred.  | 0.025  | **0.023** | **0.048** | 0.050  |
>
>
> ---
>
> 6. **“Can the author provide some relevant experiments to prove the advantages of the integrated setting to the two-stage setting that first performing imputation before generating tasks?”**
>
>     We’d like to clarify an important distinction between our architectural design and training strategy. While our framework consists of two components-a TST imputation module and a vision-based diffusion generator, the training is **not purely two-stage**.
>
>     Instead, we use the following setup:
>
>     - A short warm-up phase (30 epochs) to pre-train the TST using a reconstruction loss.
>     - Followed by joint training of both the imputation and diffusion components.
>
>     This short pre-training helps stabilize learning early on. In our experiments, training fully from scratch often led to poor reconstructions and unstable diffusion training.
>
>     To directly address your question, we compare our method to two alternatives:
>     (i) **Fully joint training** without any pre-training, and
>     (ii) **Independent two-stage training** (no joint optimization).
>
>     As shown in the Tab 2. below, our method outperforms both in discriminative score on both datasets, highlighting the value of our **integrated, pre-trained approach**.
>
>     **Table 2: Avg. discriminative scores @ 30/50/70% drop on Energy & Stock (Len=768).**
>
>     | Model                    | Energy | Stock |
>     |--------------------------|--------|--------|
>     | Joint w/o pretrain       | 0.278  | 0.076  |
>     | Two-stage only           | 0.404  | 0.122  |
>     | **Joint + pretrain (Ours)** | **0.222** | **0.024** |

---

> > ### Comment · Reviewer_KWFu · 2025-08-02
> >
> > I have read the author's response, but there are still some minor issues.
> >
> > (1) The author mentioned that when there are no missing values, it will automatically become the same architecture as ImagenTime. May I ask how it is detected? Does the sample have a NaN value directly?
> >
> > (2) The author said that their method can still be applied to the second setting I mentioned. But I think it may be difficult. Because in this type of task, our available data is usually very limited, and we need to use methods such as NeuralODE rather than purely data driven to perform it. I understand that time may be tight, but can you provide some simple experiments if possible? Because I believe that the second setting will be more practical than the first.

---

> > > ### Author Response · Authors · 2025-08-03
> > >
> > > 1. **"(1) The author mentioned that when there are no missing values, it will automatically become the same architecture as ImagenTime. May I ask how it is detected? Does the sample have a NaN value directly?"**
> > >
> > >     Yes — before passing the data through the TST module, we construct a binary observation mask based on the training data. This mask marks which time steps are observed and which are missing (i.e., contain `NaN`). If no `NaN` values are present in the sample, the mask becomes an all-ones tensor.
> > >
> > >     In this case:
> > >
> > > - The TST component becomes unnecessary and is skipped entirely — both during the warm-up phase and during joint training — since no imputation is needed.
> > > - The time series is passed directly to our improved delay embedding image transformation, which builds on ImagenTime's original transform and yields slightly better results (as we showed on the ablation study in our paper - Tab 7. [left])
> > > - The masking mechanism becomes a no-op (identity), as there are no missing values to ignore.
> > > - The diffusion model computes its loss over the full image, just as in the original ImagenTime formulation.
> > >
> > >     Thus, when the input sequence is fully observed, our architecture automatically reduces to ImagenTime — both in structure and behavior. This ensures that our method is a strict generalization of ImagenTime: it gracefully handles irregular data when needed, while maintaining the same (or slightly improved-due to our improved delay embedding transformation) performance in the fully-observed case.

---

> > > > ### Author Response · Authors · 2025-08-03
> > > >
> > > > 2. **"(2) The author said that their method can still be applied to the second setting I mentioned. But I think it may be difficult. Because in this type of task, our available data is usually very limited, and we need to use methods such as NeuralODE rather than purely data driven to perform it. I understand that time may be tight, but can you provide some simple experiments if possible? Because I believe that the second setting will be more practical than the first."**
> > > >
> > > > We thank the reviewer for raising this important point and for the thoughtful follow-up. We agree that irregular sampling frequency is a prevalent and practical challenge, especially in real-world settings with limited data. Below, we address your concerns and provide clarification on how our method accommodates this scenario, along with a discussion of design choices and comparisons to Neural ODE-based approaches.
> > > >
> > > > ### On Neural ODE-Based Methods and Their Limitations
> > > > Our work is motivated in part by addressing several limitations observed in Neural ODE-based methods such as NCDEs:
> > > >
> > > > - These methods are often computationally intensive, both in training time and memory consumption (as we showed on Tab. 2 on vxoC response).
> > > > - They are less scalable to longer sequences or larger datasets, often requiring truncation or downsampling to remain tractable.
> > > > - Their expressiveness can be limited in capturing complex or non-smooth temporal patterns, especially under high rates of missingness.
> > > >
> > > > ### Comparisons to NeuralODE-Based Baselines
> > > > Both KoVAE and GT-GAN are based on NCDEs, which themselves extend NeuralODE-based models such as ODE-RNN and GRU-ODE. Since NCDEs are known to outperform these earlier methods, our comparisons against KoVAE and GT-GAN already cover and surpass standard NeuralODE-based baselines.

---

> > > > > ### Author Response · Authors · 2025-08-03
> > > > >
> > > > > ### High Missingness Simulates Irregular Sampling
> > > > > To address the reviewer's concern more directly, we conducted additional experiments using NCDE (an extension of NeuralODE) under two input regimes:
> > > > >
> > > > > - **Setup 1: Regular grid with `NaNs`.**
> > > > >   This is the standard setup used throughout our paper and in prior works such as KoVAE and GT-GAN. We randomly replace $x\%$ of the values with `NaNs`, simulating missingness on a fixed grid. That is, we first define a uniform time grid of fixed resolution, and then explicitly mark missing values as `NaNs` at time points where no observation is available. This effectively mimics data sampled at irregular intervals, projected onto a uniform grid. For example, if the original timestamps are $t = \\{1, 3, 6, 12, 20\\}$ and we align them to a regular grid over $t = 1$ to $t = 24$, then over 80% of the entries are naturally missing and set to `NaN`. This setup is compatible with all models discussed in our paper, including TST. Importantly, when the missingness ratio is very high (e.g., $>80\%$), this setup effectively simulates data that was originally sampled irregularly. Therefore, evaluating models under Setup 1 with high missingness offers a fair and meaningful comparison to models operating under Setup 2, and directly reflects the reviewer's intended scenario.
> > > > >
> > > > > - **Setup 2: Irregular timestamps.**
> > > > >   In this setup—corresponding exactly to the setting requested by the reviewer—x% of the entries are removed entirely (without replacing with NaNs), and the remaining observations are treated as irregularly sampled inputs with continuous timestamps. This format does not involve projecting the data onto a regular grid or inserting `NaNs`, and is only compatible with ODE-based models such as NCDE.
> > > > >
> > > > > To evaluate the effectiveness of both setups, we implemented several variants using NCDE as the imputation module within our framework. Specifically:
> > > > >
> > > > > - **NCDE (with NaNs + masking)** — This variant uses the standard setup (Setup 1) employed throughout our paper, where $x\%$ of values are replaced with `NaNs`. Here, NCDE replaces TST as the imputation module, and loss masking is applied as in our original method.
> > > > >
> > > > > - **NCDE (irregular timestamps + masking)** — This variant uses the reviewer-requested Setup 2 with irregular timestamps, no `NaNs`, and NCDE as the imputation module. As in our original method, we apply loss masking to prevent the model from being penalized on potentially unreliable imputations. This setup was not evaluated in the main paper.
> > > > >
> > > > > - **NCDE (with NaNs, no masking)** — Same as the first variant, but without masking the loss. This helps demonstrate that when a large portion of values is missing, relying on potentially inaccurate imputations degrades performance, as errors propagate into the generative model.
> > > > >
> > > > > - **NCDE (irregular timestamps, no masking)** — Same as the second variant, but again without loss masking.
> > > > >
> > > > > - **TST (Ours)** — Our original method, using TST as the imputer, Setup 1, and masking in the loss.
> > > > >
> > > > > The results in Table below lead to several important conclusions:
> > > > >
> > > > > - Our original method achieves competitive performance with the reviewer's suggested setup.
> > > > > - We observe minimal difference in performance between NCDE trained with `NaNs` and NCDE trained with irregular timestamps.
> > > > > - Crucially, variants that do not apply loss masking show substantial degradation. As missingness increases, the imputation model becomes less accurate, and treating its outputs as ground truth introduces harmful supervision. Our method avoids this by masking imputed values from the loss while still using them as contextual input. This masking-based treatment of imputed values is the core novelty of our approach. It enables robust generative modeling even in high-missingness or irregularly sampled settings.
> > > > >
> > > > > Finally, we emphasize that the strength of our method lies in its modularity: both the imputation module and generative backbone can be easily swapped. Our primary contribution is the masking-based loss formulation combined with natural neighbor selection, rather than dependence on a specific imputer or generator. This is consistently supported by all of our ablation results in the paper and in the rebuttal.

---

> ### Author Response · Authors · 2025-08-03
>
> ### Discriminative scores (lower is better) for Stock dataset with 80% and 90% missingness:
>
> | **Model Variant**                                               | **80% Missing** | **90% Missing** |
> |------------------------------------------------------------------|------------------|------------------|
> | Ours - TST, ImagenTime,  Masking Loss (Setup 1)                  | **0.001**        | 0.025            |
> | NCDE with NaNs, ImagenTime, Masking Loss (Setup 1)               | 0.032            | 0.023            |
> | NCDE Irregular Frequently, ImagenTime, Masking Loss (Setup 2)    | 0.022            | **0.019**        |
> | NCDE with NaNs, ImagenTime, w.o Masking Loss (Setup 1)           | 0.158            | 0.151            |
> | NCDE Irregular Frequently, ImagenTime, w.o Masking Loss (Setup 2)| 0.165            | 0.177            |

---

> > ### Comment · Reviewer_KWFu · 2025-08-05
> >
> > Thanks for the response. I am little bit confused by the experiment and just to make sure I understand correctly, I wonder whether the experiment provided by you is that you trained your model in setup 1 on the train dataset, compared it with the model that trained on the setup 2? And then are the test datasets the same among them?

---

> > > ### Author Response · Authors · 2025-08-05
> > >
> > > Yes, exactly. In the table we added to our previous response, we trained five different models - three of them using Setup 1, and two using Setup 2.  **All models were tested on the exact same test dataset**.

---

> > > > ### Comment · Reviewer_KWFu · 2025-08-05
> > > >
> > > > Thanks for the response and I will consider updating the score accordingly. One more question is that, given this situation, I wonder whether it means the framework of the paper are not applicable to the setup 2 of training datasets? Otherwise this comparison is strange to me that you do not use the same setup of training datasets.

---

> > > > > ### Author Response · Authors · 2025-08-05
> > > > >
> > > > > The method we proposed in the paper is designed to work with TST as the imputation module. However, as we mentioned in our previous comments, TST does not support Setup 2 directly, since it requires missing values to be explicitly marked as NaNs. Therefore, our framework constructs a uniform grid and inserts NaNs at the desired locations to indicate missingness. In this sense, there is no difference in our framework between Setup 1 and Setup 2.
> > > > > In the experiments we presented, we showed that our approach performs well even under extreme missingness rates (e.g., 80% and 90%). We believe that such high missingness effectively simulates the scenario described by the reviewer, where data is irregularly sampled over time.
> > > > > Finally, if the user prefers to use Setup 2 exactly (namely, with no NaNs), this is also supported by our framework. The only required change is to switch the imputation module from TST to NCDE (or any other Neural ODE-based method), and avoid inserting NaNs altogether.

---

> > > > > > ### Comment · Reviewer_KWFu · 2025-08-05
> > > > > >
> > > > > > Thanks for the response and now since the authors addressed most of the coners, I will consider updating the score accordingly.

---

### Official Review · Reviewer_zmvt · 2025-06-30

**Clarity:** 3
**Significance:** 2
**Originality:** 2
**Rating:** 4
**Confidence:** 1

**Summary:**

This paper proposes a two-step framework for generating regular time series from irregularly sampled data. The proposed approach first completes the irregular time series using a Time Series Transformer (TST), and then applies a vision-based diffusion model (ImagenTime) with masking to generate realistic sequences. The method is designed to balance between information completion and robustness to errors in imputation. Extensive empirical results are provided, demonstrating improved performance over prior work like KoVAE and GT-GAN.

**Questions:**

- How does the proposed method compare to recent diffusion-based imputation methods like CSDI if used as a first step instead of TST?

- Did the authors consider architectures where masking is applied within the diffusion backbone (e.g., sparse convolutions or attention masks) rather than only at the loss level?

**Ethical Concerns:**

["NO or VERY MINOR ethics concerns only"]

**Final Justification:**

I would like to raise my score to 4 since the authors make hard works in responding most of my questions and most of their answers are satisfying. However, frankly speaking, my main concern to this paper is still not adequately addressed, which is the novelty. I still feel that this paper's main content is more like engineering trick rather than foundational research. If other reviewers or the Area Chair have any concerns and would like to reject this paper, I will not oppose.

**Limitations:**

Yes

**Quality:**

3

**Strengths And Weaknesses:**

**Strengths**

- The idea of combining time series completion with masking in a diffusion-based generative framework is intuitive and practically useful.

- The proposed approach shows strong empirical results across a wide range of datasets, sequence lengths, and missing rates.

- The paper is well-organized and clearly written, with helpful illustrations (e.g., Fig. 1, Fig. 2) to convey the method and its motivation.

**Weakness**

- Lack of Novelty in Core Technique: While the proposed method achieves strong results, it largely combines two well-established components (TST for completion and ImagenTime for diffusion generation) with a fairly straightforward masking trick. The main idea of using TST to impute and then applying ImagenTime with masking is more of an engineering integration than a fundamentally new modeling contribution. There is no clear technical novelty in the diffusion formulation or architectural design.

- Unclear Justification of Design Choices: The use of TST as the completion module is not well-justified beyond empirical comparison. Why is TST more appropriate than other imputation strategies, such as Gaussian processes, diffusion-based imputation (e.g., CSDI), or continuous-time models? The proposed "natural neighborhood" argument seems loosely motivated. While the visual inspection of kernel weights (Fig. 1) is illustrative, it lacks rigorous theoretical or quantitative support.

---

> ### Author Rebuttal · Authors · 2025-07-30
>
> ## Response to Reviewer zmvt
>
> We appreciate the reviewer's feedback and acknowledge the valuable points raised regarding our work. Below are our detailed responses to the comments and suggestions:
>
> ---
>
> 1. **“Lack of Novelty in Core Technique: While the proposed method achieves strong results, it largely combines two well-established components... There is no clear technical novelty...”**
>
>     We understand the concern: the components we use are indeed well-established. However, the core contribution of our work lies not in the tools themselves, but in identifying a key failure point in prior approaches and resolving it with a simple, principled solution.
>
>     Prior methods (KoVAE and GT-GAN) impute missing values and **treat them as originating from the true distribution** during training. We observed that this leads to a significant **drop in generation quality** compared to models trained on fully observed data, a gap caused by **errors in the imputed values propagating through the model**. In practice, it prevents the diffusion from learning the true distribution and thus harms its generation and generalization ability.
>
>
>     One of our main contributions is to break this assumption explicitly: **the model sees the imputed values but is never trained to predict them**, unlike KoVAE and GT-GAN. This decoupling of context and supervision is subtle but essential. While it may appear simple, it highlights our novel observation and understanding of the problem, addressing a critical issue in generating content from partial information. Specifically, when missing values are mapped to zeros in the input image, they create “unnatural” neighborhoods that mix valid and invalid information. This can pose challenges for diffusion backbones, particularly U-Nets with convolutional layers, where the kernels are not inherently masked and may inadvertently propagate errors from these artificial neighborhoods. Our approach resolves this by ensuring that the model receives coherent local structure (via imputed neighbors) while ignoring them in the loss, preventing both the propagation of errors and the absence of spatial context.
>
>     This masking strategy is not just an engineering tweak; it’s a deliberate and non-trivial change to the training objective, grounded in deep problem understanding, architectural reasoning, and empirical results. As shown in Tab. 1 below (Tab. 6 in the paper), we highlight how crucial the combination of our masked loss with natural neighbors truly is. The significant performance gains are *not due to a stronger generative backbone or superior imputation**, but stem directly from this synergy: realistic, coherent inputs paired with loss computed only on observed values. On average across all sequence lengths and missing rates, our method improves the discriminative score by 64.6\% on Stocks and 56.3\% on Energy compared to the same pipeline without masking.
>
>     To summarize, we view the novelty as lying in getting the training dynamics right for irregular time series and in identifying the failure mode that arises when imputed data is naively treated as supervision.
>
>     **Table 1: Averaged Discriminative Scores (30%, 50%, 70% drop) across Sequence Lengths (24, 96, 768)**
>
>     | Model                 | Energy 30% | Stock 30% | Energy 50% | Stock 50% | Energy 70% | Stock 70% |
>     |----------------------|-------------|------------|-------------|------------|-------------|------------|
>     | KoVAE + TST          | 0.340       | 0.173      | 0.360       | 0.176      | 0.403       | 0.139      |
>     | TimeAutoDiff + TST   | 0.297       | 0.103      | 0.333       | 0.102      | 0.465       | 0.420      |
>     | TransFusion + TST    | 0.291       | 0.082      | 0.336       | 0.092      | 0.439       | 0.104      |
>     | Ours (Mask Only)     | 0.361       | 0.170      | 0.347       | 0.294      | 0.429       | 0.372      |
>     | Ours (Without Mask)  | 0.308       | 0.028      | 0.379       | 0.073      | 0.420       | 0.066      |
>     | **Ours**             | **0.116**   | **0.014**  | **0.154**   | **0.019**  | **0.217**   | **0.014**  |
>
> ---
>
> 2. **“Unclear Justification of Design Choices: The use of TST as the completion module is not well-justified beyond empirical comparison. Why is TST more appropriate...?”**
>
>     Previous methods (KoVAE, GT-GAN) used NCDE for imputation. While accurate, NCDE introduces significant practical limitations: it is slow, does not scale to long sequences, and often becomes infeasible on high-dimensional datasets due to the need to compute cubic splines for each channel.
>
>     To address this, we replaced NCDE with TST, which is l**ightweight and scalable**. We then compared TST to various imputation methods, including CSDI, NCDE, GRU-D, linear and polynomial interpolation (**refer to Table 1 in the vxoC reviewer's response**). While TST performed slightly better in our setting (50% drop, length 24), **we do not claim that TST is an optimal imputation tool**.
>
>     In particular, we would like to emphasize that **TST is not the only strategy capable of producing natural neighborhoods**. Strong tools such as NCDE and CSDI also generate smooth, realistic completions that lead to good generative results in our framework. Our choice of TST was not due to performance alone, but primarily due to its efficiency and scalability, both during training and inference.
>
>     To justify this design decision, we added a timing benchmark comparing TST, NCDE, and CSDI across multiple sequence lengths and datasets (**refer to Table 2 in the vxoC reviewer's response**). While all three provide high-quality imputations when combined with our masking approach, TST is consistently faster, often by an order of magnitude, and remains tractable even at length 768, where other methods become extremely slow or fail entirely.
>
>     Furthermore, we note that our method is modular by design: the imputation and diffusion components can be replaced independently in future work, making the framework flexible and adaptable to a wide range of architectures and domains.
>
> ---
>
> 3. **“The proposed "natural neighborhood" argument seems loosely motivated. While the visual inspection of kernel weights (Fig. 1) is illustrative, it lacks rigorous theoretical or quantitative support?”**
>
>     We would like to clarify that the “natural neighborhood” argument is not solely based on visual inspection. In Figure 1, we already report quantitative loss metrics alongside the visualizations, demonstrating a clear improvement for the proposed kernel compared to alternatives. These improved loss values directly support the argument for natural neighborhoods, providing quantitative support in a controlled setup as well as in our extensive empirical evaluation.
>
>     Additionally, we further extended this analysis during the rebuttal phase on the Stocks dataset (please see our response to Q4. and Q5. to Reviewer vxoC), where we observed consistent performance gains, reinforcing the validity of the approach. We will revise the manuscript to make this clearer.
>
> ---
>
> 4. **“How does the proposed method compare to recent diffusion-based imputation methods like CSDI if used as a first step instead of TST?”**
>
>     We include CSDI in our ablation (**see Tab. 1 in the vxoC reviewer's response**). TST performs slightly better in our setting, but the gap is small. It is likely that in other settings, CSDI could work just as well or better; we make no claim that TST is optimal across the board. Nevertheless, we find TST to be highly efficient in comparison to other baselines, including CSDI, motivating its adoption (**Tab. 1 in vxoC**).
>
> ---
>
> 5. **“Did the authors consider architectures where masking is applied within the diffusion backbone rather than only at the loss level?**
>
>     Yes, we considered this direction and explored two preliminary variants: (1) modifying the U-Net to apply sparse masking within convolutional layers, and (2) using attention masks within self-attention layers to ignore missing pixels during attention weight computation.
>
>     We found that both approaches led to lower performance compared to our current strategy, where the diffusion model observes all imputed values as context, but the loss is computed only over the originally observed entries. Quantitative results for this comparison appear in Tab. 2 below.
>
>     Our hypothesis is that fully excluding missing pixels from the feature computation may limit the model’s ability to leverage useful local structure provided by the imputation module. In contrast, our method retains this context while avoiding error propagation — that is, the risk of reinforcing mistakes in imputed values by treating them as ground truth during training.
>
>     We also emphasize that implementing masking within the backbone typically requires architecture-specific changes — such as custom handling of sparse kernels or attention masks — which limits generality. Our approach, in contrast, is architecture-agnostic and can be applied directly to any diffusion backbone, including convolutional and transformer-based models.
>
>     That said, we agree this is an interesting direction. While our initial experiments did not show an advantage, we believe further research may uncover improved implementations that combine the strengths of both strategies.
>
>     **Table 2: Discriminative and predictive scores for different masking strategies on Energy and Stock datasets (sequence length 24, 50% missing rate). Lower is better.**
>
>     | Masking Strategy      | Disc. Energy | Disc. Stock | Pred. Energy | Pred. Stock |
>     |-----------------------|--------------|-------------|---------------|--------------|
>     | Sparse convolutions   | 0.425        | 0.291       | 0.455         | 0.076        |
>     | Attention masking     | 0.375        | 0.191       | 0.405         | 0.063        |
>     | **Ours**              | **0.065**    | **0.007**   | **0.047**     | **0.012**    |
>
>
> ---

---

> > ### Author Response · Authors · 2025-08-03
> >
> > We would like to gently remind you that we have submitted our rebuttal, which directly addresses the key concerns you raised.
> > If anything remains unclear, please let us know — your feedback is highly valued.
> > Thank you again for reviewing our work.

---

> > > ### Comment · Reviewer_zmvt · 2025-08-05
> > >
> > > I would like to raise my score since the authors make hard works in responding most of my questions and some of their answers are satisfying. However, frankly speaking, my main concern to this paper is still not adequately addressed, which is the novelty. Thus, I will only raise my score to 4 and simultaneously decrease my confidence.

---

> ### Comment · Area_Chair_teb9 · 2025-08-05
> **Please respond to authors' rebuttal before Aug. 6 (AOE)**
>
> Dear Reviewer zmvt,
>
> This is a reminder that the author-reviewer discussion period is ending soon on Aug. 6 (AOE), and you have not yet responded to authors' rebuttal. Please read authors' rebuttal as soon as possible, and engage in any necessary discussions, and consider if you would like to update your review and score. Please at least submit the Mandatory Acknowledgement as a sign that you have completed this task.
>
> Thank you for your service in the review process.
>
> AC

---

### Official Review · Reviewer_vxoC · 2025-07-02

**Clarity:** 2
**Significance:** 2
**Originality:** 3
**Rating:** 2
**Confidence:** 3

**Summary:**

This paper introduces a novel generative framework for producing regularly sampled time series from irregular and incomplete data. Irregular sampling is a common challenge in real-world applications like healthcare and finance. While previous methods such as GT-GAN and KoVAE attempted to address this, they suffer from high computational costs. To overcome these limitations, the authors propose a two-step approach. First, they use a Time Series Transformer (TST) to complete the irregular sequences, creating more natural temporal neighborhoods. This approach benefits from structured input after completion. Extensive experiments on multiple datasets and sequence lengths demonstrate that the proposed method significantly outperforms existing approaches in both accuracy and efficiency.

**Questions:**

See Weaknesses

**Ethical Concerns:**

["NO or VERY MINOR ethics concerns only"]

**Final Justification:**

**[A1]** As can be seen from the title and the contribution section, the central contribution of this paper lies in handling missing parts in irregular time series by generating natural observations and applying masked padding. My original question was why this key contribution—i.e., the proposed padding-like method—should be treated exclusively as a solution for generative modeling. Intuitively, such a method could also be applied to other tasks involving irregular time series (e.g., classification, regression). The authors claim that irregular time series are only used for imputation in such contexts; however, as demonstrated by [1], which has more than 700 citations, classification and regression tasks are also widely recognized as important in this field.

In both the main text and the authors’ rebuttal, I find no convincing explanation as to why the proposed method is specialized for generative modeling, nor why it could not address irregular time series problems in classification or regression. This omission diminishes the contribution of the paper. Of course, if the method itself had very high novelty, it could still be significant in its own right. However, the idea of leveraging an autoencoder to interpolate given time series is not new, as shown in [2]. Thus, I believe the method’s novelty is insufficient.

The authors argue that the method is “not padding,” but in the main text (line 145) they write:

> “missing values are replaced with zeros, resulting in ‘unnatural’ pixel neighborhoods. Specifically, while zeros may occasionally occur in non-zero segments of a time series, their repeated presence is highly unlikely, leading to inconsistencies. In other words, masking is not applied at the architecture level, potentially hindering the effective learning of neural components.”

Since the paper itself discusses the method in connection with padding, I believe the authors’ rebuttal claiming that it is “not padding” is not entirely appropriate.

**[A2]** It is important for a paper to explain why the proposed method is capable of achieving good performance. In this work, I understood Figure 1 to be the main component responsible for providing such an explanation. However, the “score” (score estimation loss in score-based generative models) used in the paper is, in my opinion, insufficient to explain why the method performs well. The authors should provide an explanation of why a better estimation loss is obtained. This explanation need not be fully theoretical—an intuitive argument would suffice.

My original request was for a more intuitive explanation (e.g., visualization) of why Figure 1E is better than Figure 1D. However, the authors have again substituted such an explanation with the “score” metric. While much of deep learning research is indeed phenomenological, there should at least be an intuitive reason provided for strong performance. If that is not possible, richer and more diverse experiments should be conducted to validate the claims.

The time series domain has many cases where methods believed to work well ultimately failed to generalize. For example, when the DLinear model outperformed Transformer-based models, it was a great shock to the community. This is precisely why, in this field, it is particularly important to provide a clear explanation—and extensive empirical evidence—of why a proposed method works well.

For these reasons, I ultimately recommend rejection of this paper. That said, I respect the views of the authors, other reviewers, and the AC, and I hope my comments will help the AC in making their final decision.

**References**
[1] Kidder et al., Neural Controlled Differential Equations for Irregular Time Series
[2] Jhin et al., Learnable Path in Neural Controlled Differential Equations
[3] Zeng et al., Are Transformers Effective for Time Series Forecasting?

**Limitations:**

See Weaknesses

**Quality:**

2

**Strengths And Weaknesses:**

**Strengths**
 1. The paper addresses an interesting problem. While previous methods mainly focused on how to effectively process information after applying simple zero-padding, this paper takes a different perspective by focusing on how to perform the padding itself more appropriately.
 2. It proposes a simple yet effective method.

**Weaknesses**
 1. The core contribution of this paper lies in its novel approach to handling irregularity. To better highlight this contribution, it would be necessary to evaluate performance across a wider range of tasks such as forecasting and classification, where irregularity is also a key challenge. Moreover, it would be helpful to apply the proposed strategy to existing models that handle irregularity using zero-padding, such as GRU-D or CSDI, and assess whether any performance improvement is observed. Since the proposed method is relatively simple and reuses many existing components, demonstrating its effectiveness only through the generation task makes the contribution somewhat weak.
 2. It would be better if Figure 1 were replaced with an example more closely related to a real-world problem. The current toy example does not have a direct connection to irregularity, which may lead to confusion. Additionally, the paper should provide a more concrete explanation of why the kernel in Figure 1F is considered more consistent compared to Figure 1E.

---

> ### Author Rebuttal · Authors · 2025-07-30
>
> ## Response to Reviewer vxoC
>
> We appreciate the reviewer's feedback and acknowledge the valuable points raised regarding our work. Below are our detailed responses to the comments and suggestions:
>
> ---
>
> 1. **“While previous methods mainly focused on how to effectively process information after applying simple zero-padding, ...”**
>
>     To clarify, previous methods (KoVAE, GT-GAN) did not use zero-padding; instead, they completed the missing values using NCDE, then treated the entire sequence, observed and imputed, as fully regular and trained VAE or GAN models on it. This strong assumption can harm performance, especially when the imputed values deviate from the true data distribution.
>
> ---
>
> 2. **“... it would be necessary to evaluate performance across a wider range of tasks such as forecasting and classification ... demonstrating its effectiveness only through the generation task makes the contribution somewhat weak.”**
>
>     We focused on generative modeling because it is the standard benchmark for irregular time-series completion (e.g., KoVAE, GT-GAN), and our contribution builds directly on this foundation. To ensure broader coverage and rigor, we significantly expanded prior benchmarks, from 4 to 11 datasets, and extended the evaluation beyond the standard sequence length of 24 to include lengths of 96, 768, and ultra-long time series. We also added a novel evaluation protocol with simulated Gaussian noise at multiple levels, enabling a more rigorous assessment of model robustness to noise and irregularity. We further enhanced the evaluation by incorporating both qualitative and quantitative analyses, and added additional metrics such as FID and correlation scores, offering a more complete assessment of generation quality.
>
> ---
>
> 3. **“Moreover, it would be helpful to apply the proposed strategy to existing models that handle irregularity using zero-padding, such as GRU-D or CSDI, and assess whether any performance improvement is observed.”**
>
>     We conducted an extensive ablation study comparing various completion strategies to explore the impact of different imputation methods on generative quality. These methods include simple approaches such as Gaussian noise (GN), zero-filling, linear (LI) and polynomial interpolation (PI); and advanced learning-based approaches including GRU-D, NCDE, CSDI, and our proposed Time Series Transformer (TST) completion. As shown in Tab. 1 below (which extending Tab. 6 from the paper), when local neighborhoods are unnatural, such as with zero-filling, Gaussian noise or weak imputation methods like linear interpolation, the model struggles to generate samples that closely follow the true data distribution. Conversely, more natural completion methods (PI, GRU-D, NCDE, CSDI, TST) consistently yield strong generative performance, confirming the importance of generating natural neighborhoods without overly relying on imputation accuracy.
>
>     To further motivate our choice of TST as the completion strategy, we emphasize that methods like CSDI and NCDE depend on computationally expensive operations such as cubic spline calculation in NCDE or 1000 sampling steps, respectively, which becomes extremely slow or infeasible for long sequences or high-dimensional data due to high memory and runtime demands as can be shown in the lower table below. In conclusion, TST is a lightweight and scalable module that offers competitive or better performance with significantly improved efficiency. See Tab. 2 below for full experiment results.
>
>    **Table 1: Discriminative and predictive scores for different imputation methods on Energy and Stock datasets (sequence length 24, 50% missing rate)**
>
>    | Imputation Method             | Disc. Energy | Disc. Stock | Pred. Energy | Pred. Stock |
>    |------------------------------|--------------|-------------|--------------|-------------|
>    | Gaussian noise (→ NaN)       | 0.457        | 0.102       | 0.058        | 0.014       |
>    | Zero-filling (→ NaN)         | 0.269        | 0.158       | 0.051        | 0.014       |
>    | Linear interpolation (LI)    | 0.251        | 0.013       | 0.049        | 0.019       |
>    | Polynomial interpolation (PI)| 0.201        | 0.012       | 0.053        | 0.016       |
>    | GRU-D                         | 0.158        | 0.014       | 0.055        | 0.015       |
>    | NCDE                          | 0.102        | 0.013       | 0.058        | 0.013       |
>    | CSDI                          | 0.088        | 0.012       | 0.048        | 0.013       |
>    | **TST (Ours)**                | **0.065**    | **0.007**   | **0.047**    | **0.012**   |
>
>    **Table 2: Training and imputation time for TST, NCDE, and CSDI on RTX 3090 (batch size fixed)**
>
>    | Dataset | Seq. Len | Ours - TST Train (h) | Ours - TST Impute (s) | NCDE Train (h) | NCDE Impute (s) | CSDI Train (h) | CSDI Impute (min) |
>    |---------|----------|---------------|----------------|----------------|------------------|----------------|--------------------|
>    | Energy  | 24       | **0.67**      | **0.64**       | 2.55           | 2.4              | 1.60           | 35.71              |
>    |         | 96       | **0.80**      | **0.74**       | 7.68           | 5.6              | 5.43           | 135.29             |
>    |         | 768      | **6.67**      | **6.26**       | 60.19             | 56.0              | 84.61          | 2394.71            |
>    | Stock   | 24       | **0.10**      | **0.19**       | 0.67           | 2.0              | 0.18           | 15.40              |
>    |         | 96       | **0.13**      | **0.20**       | 2.45           | 7.2              | 0.40           | 46.78              |
>    |         | 768      | **1.10**      | **1.20**       | 59.32          | 56.0             | 3.55           | 514.08             |
>
> ---
>
> 4. **“It would be better if Figure 1 were replaced with an example more closely related to a real-world problem. The current toy example does not have a direct connection to irregularity, which may lead to confusion.”**
>
>     Following the reviewer's suggestions, we replicated the experiment using a real-world dataset, Stocks. For this experiment, we applied a larger convolutional kernel of size $6 \times 6$ and evaluated two settings: (i) filling irregularities with zeros, and (ii) filling them using natural neighbors. The results on the Stocks dataset mirrored those observed in the synthetic setting, reinforcing our hypothesis about the detrimental effect of "unnatural" neighbors.
>
>    Specifically, we trained two diffusion models:
>    (i) one that predicts the score across the entire image,
>    (ii) one that predicts only the score corresponding to valid (non-zero) values using masking.
>
>    We evaluated both models by measuring the score estimation loss on the active pixels. The results show that masking yields only a marginal improvement, with losses of **0.81** and **0.79** for setups (i) and (ii), respectively. A visual inspection confirms that the behavior is consistent with the patterns observed in the synthetic experiment. Finally, when applying our proposed approach, which avoids reliance on zero-padding and learns from semantically valid neighborhoods, we observe significantly improved performance with a much lower loss of **0.29** and more consistent kernel behavior.
>
> ---
>
> 5. **“The paper should provide a more concrete explanation of why the kernel in Figure 1F is considered more consistent compared to Figure 1E”**
>
>     We acknowledge the ambiguity in the original phrasing. Our intention was not to imply that the kernel in Figure 1F is more consistent than the one in Figure 1E, but rather that both 1E and 1F exhibit a similar level of consistency, which is notably improved compared to Figure 1D. Importantly, the kernel in Figure 1F achieves a better quantitative score than 1E, reflecting its superior performance. We will revise the sentence in the manuscript to clearly reflect this distinction.
>
> ---

---

> > ### Author Response · Authors · 2025-08-03
> >
> > We would like to gently remind you that we have submitted our rebuttal, which directly addresses the key concerns you raised.
> > If anything remains unclear, please let us know — your feedback is highly valued.
> > Thank you again for reviewing our work.

---

> > ### Comment · Reviewer_vxoC · 2025-08-05
> >
> > Thank you for your responses to my concerns.
> > However, my concerns have not been resolved yet.
> > The main contribution of this paper lies in proposing a new padding method to handle irregularity, which is not specifically tailored for generation tasks alone. Given this key contribution, I believe it is necessary to demonstrate its effectiveness on other tasks where irregularity is also a challenge, such as classification or regression problems. Additionally, although the authors mentioned several improvements related to visual outputs, there are still concerns that remain unresolved due to the lack of visual confirmation. (I understand this may be due to the inability to upload a PDF file, but the concern was not sufficiently addressed based on the provided text alone.) Considering these points, I will maintain my previous score.

---

> > > ### Author Response · Authors · 2025-08-07
> > >
> > > 1. **"The main contribution of this paper lies in proposing a new padding method to handle irregularity, which is not specifically tailored for generation tasks alone. Given this key contribution, I believe it is necessary to demonstrate its effectiveness on other tasks where irregularity is also a challenge, such as classification or regression problems."**
> > >
> > >     To begin, our method is **not a padding strategy**, nor is it designed to enhance representations for downstream tasks. Instead, our contribution addresses a core challenge in generative modeling:
> > >     How can we train a model to generate fully observed, regularly sampled time series, using only irregularly sampled data with missing values during training?
> > >      In this setting, no access to complete data is available at any train stage, including imputation. Prior works such as KoVAE and GT-GAN attempt to fill missing values and train generative models as if the imputations were ground truth—leading to error propagation when those values are inaccurate.
> > >     Our method avoids this by masking imputed regions during training (and not by padding). **We treat completed values as context but exclude them from the loss**. This encourages the model to benefit from local structure without overfitting to potentially erroneous completions. The novelty lies **not in the imputation model** (e.g., TST), but in the masking strategy and training objective that make this work (for more details about the contribution of the masking mechanism and natural neighbors - please refer to Tab. 1 in zmvt response).
> > >
> > >     We agree that tasks such as classification and forecasting are important in the broader landscape of irregular time series modeling. However, in context of irregular time series, these tasks are often used to evaluate imputation models, since high-quality imputations should yield representations that generalize well to downstream objectives. In contrast, our work does not claim to improve imputation, nor do we propose a better representation learning scheme. Our focus is strictly on generative modeling, and our design choices are tailored to this goal. Therefore, evaluation via forecasting or classification would be misaligned with our research question.
> > >     Still, to evaluate generation quality, we adopt a predictive protocol: a forecasting model is trained on the generated data and tested on real data. This assesses whether generated samples carry meaningful predictive signals. While it is not a classical forecasting downstream task, it serves as a practical proxy for measuring generation utility.
> > >     Finally, we do not dismiss the reviewer’s suggestion. We believe that adapting our method to downstream tasks such as classification or forecasting is a promising direction for future work, and our modular architecture may lend itself to such extensions. However, this would require non-trivial design adjustments, including new objectives and inference strategies. As it stands, our paper targets a well-defined and novel generative problem - and offers a general framework for learning from irregular data without relying on imputed values as supervision.

---

> ### Comment · Area_Chair_teb9 · 2025-08-05
> **Please respond to authors' rebuttal before Aug. 6 (AOE)**
>
> Dear Reviewer vxoC,
>
> This is a reminder that the author-reviewer discussion period is ending soon on Aug. 6 (AOE), and you have not yet responded to authors' rebuttal. Please read authors' rebuttal as soon as possible, and engage in any necessary discussions, and consider if you would like to update your review and score. Please at least submit the Mandatory Acknowledgement as a sign that you have completed this task.
>
> Thank you for your service in the review process.
>
> AC

---

> ### Author Response · Authors · 2025-08-07
>
> 2. **"Additionally, although the authors mentioned several improvements related to visual outputs, there are still concerns that remain unresolved due to the lack of visual confirmation. (I understand this may be due to the inability to upload a PDF file, but the concern was not sufficiently addressed based on the provided text alone.) Considering these points, I will maintain my previous score."**
>
>     We'd like to clarify the experimental setup and address any remaining concerns. Following your suggestion, we extended this experiment to real-world data to validate the robustness of our findings. Although difficult to convey fully in text, we observed the same "neighbors" phenomenon described in Section 4 in our paper.
>
>     It seems we may not have clearly communicated the visual insights from the kernel plots, so we'll briefly outline them here: Figures 1B and 1C depict two environments: one with "non-natural neighbors" [B], where the data is surrounded by zeros, and another with "natural neighbors" [C], where completions are generated by a TST-like model. The central data is identical in both.
>
>     We then train a diffusion model with a single kernel in setup B to observe what the kernel learns. By plotting the kernel's magnitude, we assess which regions it prioritizes during diffusion. In Figure 1D, we see that when trained without masking, the kernel attends more to the surrounding environment rather than the actual data. With masking, however, the kernel learns to focus on the relevant regions. This faciliate the motivation for the incorporation of masking in our method.
>
>     Next, in setup C, we replace the zeros with more realistic completions. Since masking helps focus the kernel, we ask: how well does it focus now? To evaluate this, we report the "score" that reflects the kernel's ability to estimate the diffusion process [1, 2]. We find that with natural neighbors, the score is twice as low, indicating a significantly better estimation. Figure 1E shows the resulting kernel, visually similar to 1D due to masking but with fundamentally improved estimation quality.
>
>     Lastly, we emphasize that this improved behavior is consistent with our findings on real-world data. We hope this clarifies the experiment and are happy to address any further questions.
>
> ---
>
> [1] Generative Modeling by Estimating Gradients of the Data Distribution
>
> [2] Score-Based Generative Modeling through Stochastic Differential Equations

---

### Note · Authors · 2025-08-11

Dear AC,

We appreciate the opportunity to provide these final remarks. We summarize below the main concerns and our responses.

**Evaluation Scope and Experimental Validation:** vxoC asked about downstream tasks. We explained that while imputation and representation methods often rely on them, irregular TS generative methods are evaluated by how realistic the generated data is. Our work aligns with generative benchmarks and adapting it for downstream tasks would require non trivial changes. To address concerns from vxoC and zmvt regarding imputer choise, we compared TST to GRU-D and CSDI. These comparisons (Tab. 1 vxoC) confirmed that our masked-loss strategy yields imputer-agnostic gains. We further showed (Tab. 2 vxoC) that TST is a lightweight, scalable module offering strong performance and efficiency. We addressed concerns about the synthetic experiment (Fig. 1) by replicating it on a real-world dataset (Stocks), confirming the value of natural neighbors as supervision.

**Core Methodological Contributions:** Some reviewers, including zmvt, raised questions about the contribution relative to prior work. We clarified that our method addresses a key failure mode in earlier approaches: treating imputed values as true distribution during training, which propagates errors and harms generative performance. Our proposed solution, loss masking combined with natural neighbors, is simple and effective. We showed that neither masking nor TST alone accounts for the gains, but their integration is critical (Tab. 1 zmvt).

**Robustness and Generalization:** KWFu raised concerns about the handling of irregular sampling and real-world missingness. We showed that our framework naturally handles irregular sampling via projection to a regular grid with explicit NaNs, and is robust to both random and structured (block-wise) missingness.

**Baselines and Mixed-Type Data:** ysta requested comparisons to diffusion+VAE. We compared against TimeAutoDiff, showing that our method significantly outperforms them under the same imputer and evaluation setup. We also extended our method to handle mixed-type data via learnable embeddings, showing strong results on the Traffic dataset.

In summary, we addressed all major concerns through new experiments, clarifications, and design insights. While some broader questions (e.g. extensions to downstream tasks) remain for future work, the discussion was constructive, and we hope this positive momentum carries into the final decision.

---

### Decision · Program_Chairs · 2025-09-17

**Decision:**

Accept (poster)

**Comment:**

This submission proposes a generative model for irregular time series. The method first completes missing values using a Time Series Transformer (TST) to create more natural temporal neighborhoods, and then using a vision-based diffusion model (ImagenTime) with masking to generate realistic time series. The underlying intuition of this approach is to overcome the issue of unnatural neighborhoods caused by zero-padding. Empirical results demonstrate the improved performance, and also improved training efficiency.

The reviewers and I myself agree that this is indeed an important and practical problem. The proposed method seems reasonable, and the empirical performance seems advantageous across a wide range of datasets and tasks. The authors (and the reviewers) also elaborated a lot in the rebuttal period, and more comprehensive results are provided, which also seem supportive.

Nevertheless, most reviewers have mentioned the concern of limited novelty, as the framework mainly combines existing components (time series completion via TST, and diffusion-based generation via ImagenTime) without introducing fundamentally new architectures or theoretical insights.
I do agree with this situation and also expect more exciting novel components/analysis, but I tend to believe that a comprehensive empirical study on a not-to-involved strategy that seems to work well also accounts a reasonable contribution, and could be inspiring to the community. Therefore, I'm leaning towards acceptance.